# Spatio-spectral 4D coherent ranging using a flutter-wavelength-swept laser

Dawoon Jeong[1,2], Hansol Jang[3], Min Uk Jung[1,2], Taeho Jeong[4], Hyunsoo Kim[5], Sanghyeok Yang[5], Janghyeon Lee [5] & Chang-Seok Kim [1,2] ✉

Coherent light detection and ranging (LiDAR), particularly the frequency-modulated continuous-wave LiDAR, is a robust optical imaging technology for measuring long-range distance and velocity in three dimensions (3D). We propose a spatio-spectral coherent LiDAR based on a unique wavelength-swept laser to enable both axial coherent ranging and lateral spatio-spectral beam scanning simultaneously. Instead of the conventional unidirectional wavelength-swept laser, a flutter-wavelength-swept laser (FWSL) successfully decoupled bidirectional wavelength modulation and continuous wavelength sweep, which overcame the measurable distance limited by the sampling process. The decoupled operation in FWSL enabled sequential sampling of flutter-wavelength modulation across its wide spectral bandwidth of 160 nm and, thus, allowed simultaneous distance and velocity measurement over an extended measurable distance. Herein, complete four-dimensional (4D) imaging, combining real-time 3D distance and velocity measurements, was implemented by solid-state beam scanning. An acousto-optic scanner was synchronized to facilitate the other lateral beam scanning, resulting in an optimized solid-state coherent LiDAR system. The proposed spatio-spectral coherent LiDAR system achieved high-resolution coherent ranging over long distances and real-time 4D imaging with a frame rate of 10 Hz, even in challenging environments.

Optical imaging techniques based on optical coherence are used as powerful tools in various applications, ranging from optical coherence tomography[1] for imaging on the order of several millimeters[2,3] to coherent light detection and ranging (LiDAR) for imaging on the order of several tens of meters[4–6]. Coherent LiDAR technology is gaining interest, particularly because of the increasing demand for high-quality three-dimensional (3D) imaging for remote sensing in various fields, such as autonomous driving[7,8], defense[9], security, and robotics. Coherent LiDAR, also known as frequency-modulated continuous-wave (FMCW) LiDAR[10], has recently received significant attention owing to its unique advantages compared to the conventional time-of-

flight[11–13] and amplitude-modulated continuous-wave LiDAR. Based on the coherent detection principle, coherent LiDAR is robust against external environmental disturbances[14] and ambient light interference. It simultaneously provides accurate distance and velocity information at a high axial resolution and signal-to-noise ratio (SNR) over long distances.

In addition to its strengths of coherent measurement, there are ongoing initiatives to develop high-performance coherent LiDAR systems that excel in terms of accuracy, speed, resolution, range, stability, and efficiency. Various approaches have been reported to achieve higher performance, especially in solid-state systems[15]. Flash

[1]Department of Cogno-Mechatronics Engineering, Pusan National University, Busan 46241, Korea. [2]Engineering Research Center for Color-Modulated Extra-Sensory Perception Technology, Pusan National University, Busan 46241, Korea. [3]Ground Technology Research Institute, Agency for Defense Development, Daejeon 34186, Korea. [4]Energy Device Research Team, Hyundai Motor Company, Uiwang, Gyeonggi 16082, Korea. [5]Electromagnetic Energy Materials Research Team, Hyundai Motor Company, Uiwang, Gyeonggi 16082, Korea. ✉e-mail: ckim@pusan.ac.kr

LiDAR[4,16,17] exhibit powerful performance for 3D distance and velocity detection over a long range with a compact on-chip architecture. However, it has the drawback of a limited number of pixels. The resulting narrow field of view (FOV) and consequent requirements for free-space optics indicate that the technology requires further maturation. Similarly, LiDAR systems based on optical-phased arrays (OPA)[18-20] deliver robust performance with a silicon chip architecture and offer a wider FOV without relying on a lens. However, the fabrication of an OPA involves a challenging trade-off among the FOV, SNR, and beam efficiency. Additionally, a full four-dimensional (4D) image of 3D distance and velocity using this technology has not been reported yet. Coherent LiDAR using spectral dispersion[4,21-24] is a strong alternative, especially when compared to existing silicon chip-based LiDAR systems, including flash and OPA-based LiDAR. Spatio-spectral mapping simplifies it straightforwardly to create a direct correspondence of the lateral position of the beam according to wavelength sweep, all driven by a single light source.

Given the specific features of the light source required for coherent LiDAR, in which the wavelength must be swept, it is an efficient strategy to use the broad wavelength bandwidth of the light source to achieve coherent axial ranging and lateral beam-scanning simultaneously. Okano et al.[21] first proposed a swept-source LiDAR using a combination of a wideband wavelength-swept source and diffraction grating. A high sweep rate of 10 kHz with a sweep bandwidth of 40 nm at a center wavelength of 1060 nm was achieved using a commercially available tunable vertical-cavity surface-emitting laser. During the unidirectional wavelength sweep of the source, light is spatially mapped in one lateral direction by passing through the diffraction grating. The interference signal across the entire bandwidth was measured and subsequently processed after windowing at arbitrary sampling point intervals to separate the axial coherent ranging and lateral spatial scanning. A 3D distance image of 30 × 100 pixels was obtained at an approximate single-point acquisition rate of 13.5 kHz with an axial resolution of 0.042 cm. Another 3D distance image of 45 × 200 pixels was obtained, utilizing a galvanometric scanner for the remaining lateral axis. The second image was acquired at an acquisition rate of 300 Hz with an axial resolution of 0.14 cm. This was the first demonstration of a spatio-spectral coherent LiDAR system based on a broadband wavelength-swept laser and diffractive optics, demonstrating effective and efficient operation with a simple system configuration.

Qian et al.[22] proposed a time-frequency multiplexed 3D coherent ranging system that extended the previous approach by Okano et al. They utilized a commercial akinetic all-semiconductor wavelength-swept laser with a sweep rate of 15.94 kHz and a wavelength bandwidth of 65.85 nm, centered at 1316 nm. Similarly, the interference signals throughout the wavelength bandwidth were sampled at once and then post-processed to derive both the axial distance and lateral position information. A densely sampled high-resolution spatio-spectral coherent LiDAR system was demonstrated using a compressed sampling method during the post-processing stage. It showed a 3D distance image of 475 × 400 pixels at an acquisition rate of 7.6 MHz with an axial resolution of 0.28 cm. High-resolution imaging at a fast acquisition rate, enabled by optimization at sampling point intervals, ensured powerful spatio-spectral coherent LiDAR (the specifications of the laser source and LiDAR system are listed in Supplementary Table 1).

However, there are two significant drawbacks that still need to be addressed: (1) The system cannot measure the Doppler frequency for velocity owing to the unidirectional wavelength sweep across the entire bandwidth; consequently, it can only be utilized for distance ranging. (2) The system has a short measurable distance owing to the constraints of the sampling process. In addition to the laser coherence length, the maximum measurable distance is calculated as $d_{max} = f_s cT/4B$, where $f_s$, $c$, $T$, and $B$ denote the sampling rate of the digitizer, the speed of light, the sweep period, and the sweep

bandwidth[21], respectively. Moreover, it is assumed that the beat frequency of the entire wavelength bandwidth does not exceed the response bandwidth of the avalanched balanced photodetector (BPD). Because the complete distance data were simultaneously collected and sampled along a single lateral axis, the measurable distance was significantly limited by the finite electrical bandwidth compared to the provided wavelength bandwidth. In other words, there is a significant trade-off among the measurable distance, the sweep rate, and the wavelength bandwidth. Okano et al. extended the measurable distance from 0.25 to 12 m by reducing the sweep rate and bandwidth to 300 Hz and 19 nm, respectively. However, this approach is less desirable because it results in the loss of advantages associated with spatio-spectral mapping and real-time ranging. To develop a high-performance LiDAR system suitable for real-world applications[8], the fundamental requirement of a coherent LiDAR system is the capability of 4D ranging over long distances. The subsequent requirements include high-resolution real-time imaging, precise measurements, and stable operation within a straightforward and efficient system configuration. In pursuit of these goals, it is clear that the light source plays a crucial role in the development of a spatio-spectral coherent LiDAR system. Specifically, a suitable wavelength-swept laser must have a coherence length adequate for long-distance ranging[25] as a primary requirement. In addition, the laser must offer bidirectional modulation throughout the continuous and linear wavelength sweep[26]. This capability is vital for measuring distance and velocity during spatio-spectral scanning. In this context, data-efficient sampling[27,28] of the interference signal across the entire wavelength bandwidth is essential for extending the measurable distance. Importantly, all these requirements must be fulfilled without compromising the wavelength bandwidth and sweep rate.

In this study, we introduce an innovative laser source, a flutter-wavelength-swept laser (FWSL), and a solid-state spatio-spectral coherent LiDAR system that utilizes the FWSL in combination with an acousto-optic deflector (AOD). The proposed FWSL simultaneously offers bidirectional wavelength modulation at the pm-level and wavelength sweep at the hundred nm-level each enabling axial coherent ranging and spatio-spectral mapping, respectively. We achieved independent mechanisms for the wavelength modulation in a fluttering shape (bidirectional modulation), and the wavelength sweep across the entire wavelength bandwidth. This decoupling, which occurred at the laser source level, separated axial ranging and spatio-spectral scanning functionalities in the resulting LiDAR system. The advancement allows for concurrent coherent 4D measurements throughout spatio-spectral mapping without any conflicts by instantly and sequentially sampling the flutter-wavelength modulation segment during the continuous wavelength sweep. This approach effectively resolved the measurable distance limitations of previous sampling methods. The proposed FWSL offers a long coherence length of 1.2 km, a wide bandwidth of 160 nm centered at 1535 nm, and a rapid sweep rate of 100 kHz per single acquisition. To complement the spatio-spectral scanning along the horizontal axis achieved with diffraction grating, an AOD[29] was used as a non-mechanical scanner for the vertical axis. This study showcases the potential of a flawless solid-state coherent LiDAR system, an achievement not realized in prior studies utilizing synchronized triple-driving configurations.

We simultaneously obtained 3D distance and velocity images using coherent ranging and solid-state beam scanning. A theoretical maximum measurable distance of 264 m was achieved through the sequential sampling of the flutter-wavelength modulation segment, and successful results of coherent ranging were shown in order of several tens of meters. In conclusion, we successfully acquired high-resolution 4D images of miniaturized targets in real-world settings, even in a challenging environment, achieving a real-time frame rate in dimensions of 200 × 45 pixels with the robust and efficient solid-state spatio-spectral coherent LiDAR system.

## Results

### FWSL-based solid-state coherent LiDAR system

The architecture of the proposed solid-state coherent LiDAR system based on the FWSL is shown in Fig. 1. The system comprised an FWSL, an external cavity-based wavelength-swept laser, an interferometer, and an optical system that included a solid-state scanner. The entire system was designed in a synchronized triple all-electrical driving configuration of an electro-optic phase modulator (EOPM), AOD, and wavelength selector (WS) to obtain the axial, vertical, and horizontal information, respectively. As shown on the left side of Fig. 1a, the FWSL contained an EOPM within a cavity comprised of a semiconductor optical amplifier in the 1550 nm band to the WS. The WS determined the wavelength of the light to be lased, and the EOPM modulated the selected wavelength. The EOPM was controlled by an electric signal from an arbitrary function generator (AFG) that enabled bidirectional modulation with a freely adjustable modulation signal. In addition, the WS was controlled by an electrical signal from the AFG, allowing the continuous and linear wavelength sweep over the entire bandwidth. The EOPM and WS were triggered by each other to ensure robust flutter-wavelength modulation for each fixed wavelength across the entire wavelength bandwidth. Wavelength modulation via EOPM and wavelength sweep via WS were operated simultaneously yet independently, which decoupled the axial coherent ranging and lateral spatio-spectral mapping.

Light from the FWSL was mapped in free space using a solid-state beam scanner: AOD for acousto-optical scanning along the vertical axis

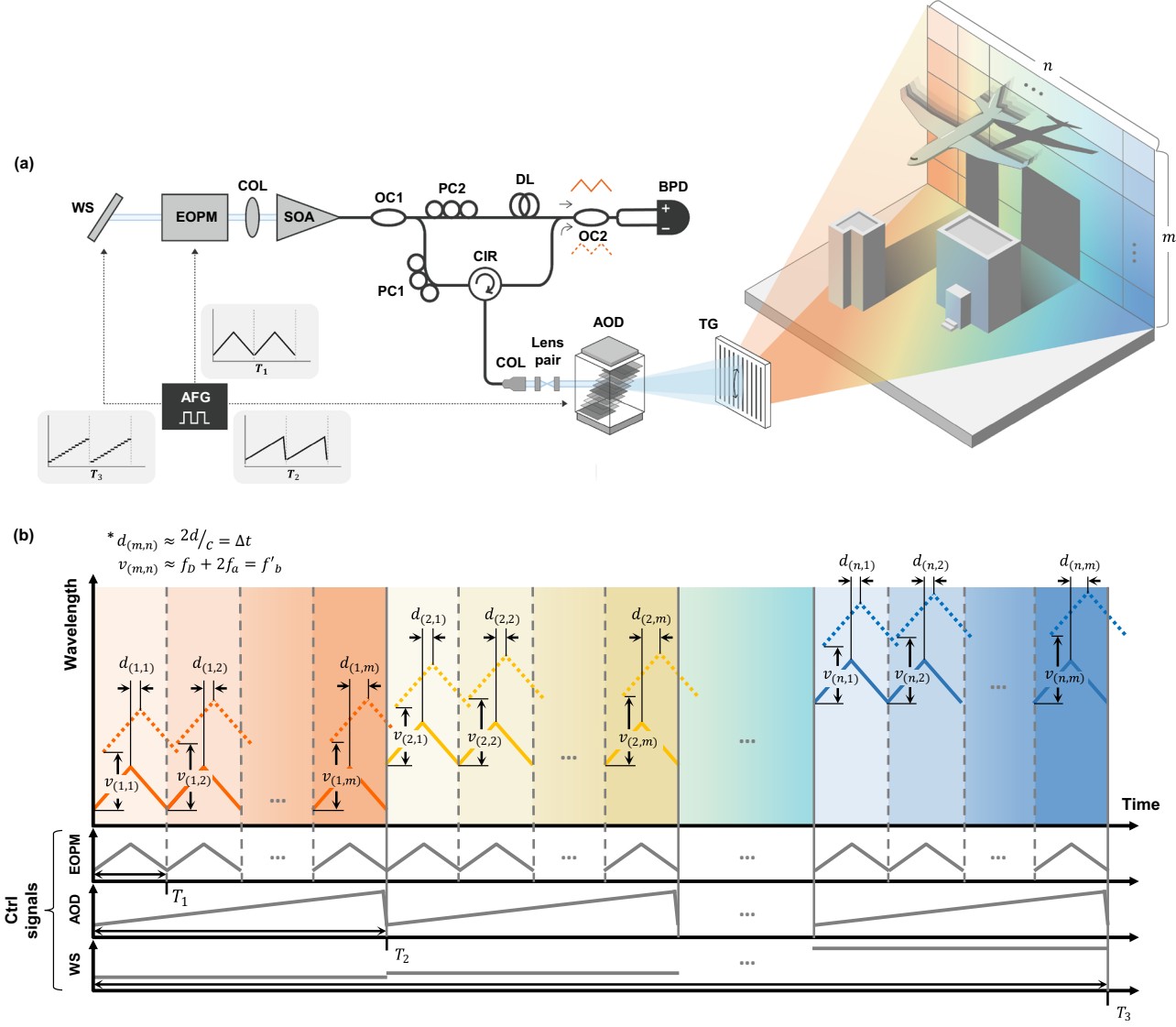

**Fig. 1 | Design and the coherent ranging principle of the solid-state coherent light detection and ranging (LiDAR) system based on flutter-wavelength-swept laser (FWSL). a** Design of the FWSL-based solid-state coherent LiDAR system. The system comprises an FWSL from a wavelength selector (WS) to a semiconductor optical amplifier (SOA), a fiber-based Mach–Zehnder interferometer, a solid-state beam scanner of an acousto-optic deflector (AOD) in the vertical direction, and transmission grating (TG) in the horizontal direction. The optical paths in free space and the fiber are shown as sky-blue and black solid lines, respectively. Electric control signals (ctrl signals) with periods of $T_1$, $T_2$, and $T_3$ are indicated by black dotted arrows. EOPM, AFG, OC, PC, CIR, DL, COL, and BPD represent electro-optic phase modulator, arbitrary function generator, optical coupler, polarization controller, circulator, delay line, collimator, and balanced photodetector, respectively. **b** 4D coherent ranging principle using the proposed coherent LiDAR system. Each ctrl signal from the AFG modulates the EOPM, AOD, and WS for coherent ranging, vertical, and horizontal scanning, respectively. $d_{(m,n)}$ and $v_{(m,n)}$ represent the distance and velocity measured at every axial point. $d_{(m,n)} \approx 2d/c = \Delta t$ and $v_{(m,n)} \approx 2f_a + f_D = f'_b$, where $d$ is the real distance of the target, $c$ is the speed of light, $\Delta t$ is the time delay owing to the distance, $f_a$ is the acoustic frequency, and $f_D$ is the Doppler frequency.

and transmission grating (TG) for spatio-spectral scanning along the horizontal axis. The light passed through the AOD, a device that deflects light according to the acoustic frequency of an applied electrical signal, was then dispersed by passing through the TG. In this study, a two-dimensional (2D) raster scan was performed with the acousto-optical and spatio-spectral axes as the fast and slow axes, respectively. An overall 4D ranging principle using 2D solid-state scanning for simultaneous distance and velocity measurement based on coherent detection is summarized in Fig. 1b. First, the wavelength was modulated by the EOPM according to a triangular waveform with period $T_1$ for the coherent detection of a single axial point. Next, the same flutter-wavelength modulation was performed for the number of vertical pixels by scanning the beam in the vertical direction through the AOD. The vertical axis was scanned along the signal with period $T_2$ of the ramp waveform applied to the AOD. Finally, the same process was performed for the number of horizontal pixels according to a step waveform with period $T_3$, scanning the beam along the horizontal axis, where the wavelength was continuously swept through the WS. As a result, a 4D coherent image of $n \times m$ pixels (H × V) was obtained by splitting the entire wavelength bandwidth into $n$ and repeating the flutter-wavelength modulation at each wavelength $m$ times along the acousto-optical axis.

The FWSL and optical system, which included a solid-state scanner, were interconnected with a fiber-based Mach–Zehnder interferometer for coherent detection. The light emitted from the FWSL passed through an optical coupler and was split into reference and sample arms. The light propagating through the sample arm was mapped into a 2D space by passing through the solid-state scanner. The reflected light returned to the reverse order and was coupled back through the circulator to interfere with the light of the reference arm at the other optical coupler. The interference signal was detected using a BPD and processed using a digitizer. Detailed data acquisition and processing procedures are described in the Methods section.

## Coherent ranging and velocity measuring principle

The EOPM, AOD, and WS function for coherent ranging in the axial direction and beam scanning in the vertical and horizontal directions, respectively. They were triggered by each other via AFG for electrically synchronized operations. The EOPM was modulated with a triangular waveform of a fixed period, $T_1$, of 10 µs for the flutter-wavelength modulation. The wavelength modulation bandwidth was measured as 0.71 GHz (5.5 pm at the center wavelength of 1535 nm) with respect to the applied modulation signal (Fig. 2b). The Methods and Supplementary Note 7 present the detailed measurement process and analysis. The R-squared values for each linear variable period for the up-chirp and down-chirp were 0.99997 and 0.99987, respectively. The high linearity enables accurate coherent ranging without an additional assistance of a k-linear process[6]. In coherent ranging, the distance ($d$), and velocity ($v$) are calculated using the following equations[23,30]:

$$d = \frac{cT}{2B}f_b = \frac{cT}{2B}\frac{f_{b,up}+f_{b,down}}{2}, \tag{1}$$

$$v = \frac{\lambda}{2}f_D = \frac{\lambda}{2}\frac{f_{b,up}-f_{b,down}}{2}, \tag{2}$$

where $B$ is the wavelength modulation bandwidth, $c$ is the speed of light, $T$ is the linear wavelength modulation period, and $\lambda$ is the center wavelength. $f_b, f_{b,up}, f_{b,down}$, and $f_D$ represent the actual beat frequency for distance, the detected beat frequency during the up-chirp and down-chirp, and the Doppler frequency for velocity, respectively. The theoretical axial resolutions for distance and velocity[23,30] determined by the FWSL were 0.21 m and 0.15 m/s, respectively, according to $\Delta d = c/2B$ and $\Delta v = \lambda/2T$.

The AOD was operated using an electric signal with a ramp waveform of period $T_2$. The AOD exhibited a frequency bandwidth of 18 MHz with respect to a center acoustic frequency of 52 MHz at a working wavelength of 1550 nm. The scan angle corresponding to the frequency bandwidth was ~3.5°, which was the vertical FOV of the system. However, the acoustic frequency applied to the AOD affects not only the deflection angle of the incoming light but also its wavelength (i.e., optical frequency). The optical frequency of the light passing through the AOD shifts in the positive direction by the amount of that acoustic frequency[31,32], depending on the acoustic frequency applied to the AOD. This acoustic frequency shift affected the beat frequency, which is the optical frequency difference between the sample and reference arms. The acoustic frequency shift depending on the chirp shape differs by the following equations: $f'_{b,up} = 2f_a - f_b$ and $f'_{b,down} = 2f_a + f_b$, where $f'_{b,up}$ and $f'_{b,down}$ represent the beat frequency during the up-chirp and down-chirp, respectively. Two times the acoustic frequency of $f_a$ in the equations is caused by the round trip of light. The acoustic frequency shift in the beat frequency is shown in Fig. 2c, with the fast Fourier transform (FFT) result of the interference signal from a coherent range against a target at 20 m. The detected beat frequencies during up-chirp and down-chirp, the two different peaks in red and blue, respectively, are distributed in opposite directions with respect to the center of 104 MHz (the black dotted line), which corresponds to $2f_a$, following the relationships above. According to this acoustic frequency shift, the actual beat and Doppler frequencies are related to the measured beat frequencies in our LiDAR system, as follows:

$$f_b = \frac{\left| f'_{b,up} - f'_{b,down} \right|}{2} \tag{3}$$

$$f_D = \frac{f'_{b,up} + f'_{b,down}}{2} - 2f_a \tag{4}$$

The compensation process for the acoustic frequency required to decompose the actual velocity information, is described in the Methods section.

The WS was operated according to the control signal of a step waveform with period $T_3$. A wavelength bandwidth of 160 nm with a center wavelength of 1535 nm was achieved by WS, as shown in Fig. 2e (a detailed characterization procedure is provided in the Methods section). The horizontal FOV determined by the TG for the entire wavelength bandwidth was ~14.5°. The linewidth at the center wavelength of 1535 nm was measured at 81 kHz, which corresponds to a theoretical coherence length of 1.2 km (Supplementary Fig. 9). As mentioned previously, each signal was operated while being triggered in accordance with the scanning process. Coherent measurements were performed by appropriately setting the variable conditions of AOD and WS (spatial scanning conditions) depending on the targeted scenes, while the axial scanning condition by EOPM was kept fixed.

## Characterization of coherent LiDAR system

Coherent measurements were performed on a rotating disk with a retroreflective tape applied to its surface to characterize the proposed FWSL-based LiDAR system. The experimental setup is shown in Fig. 2a. A disk with a radius of 0.125 m was positioned 1 m behind the TG, rotating at a speed of 770 rpm. A 3D velocity image of 100 × 20 pixels (H × V) was obtained for the central area of the disk, where the velocity changed from positive to negative owing to the receding and approaching parts against the beam propagation, respectively. The center of the disk was aligned perpendicular to the incident beam when the AOD and WS were set to the center acoustic frequency and

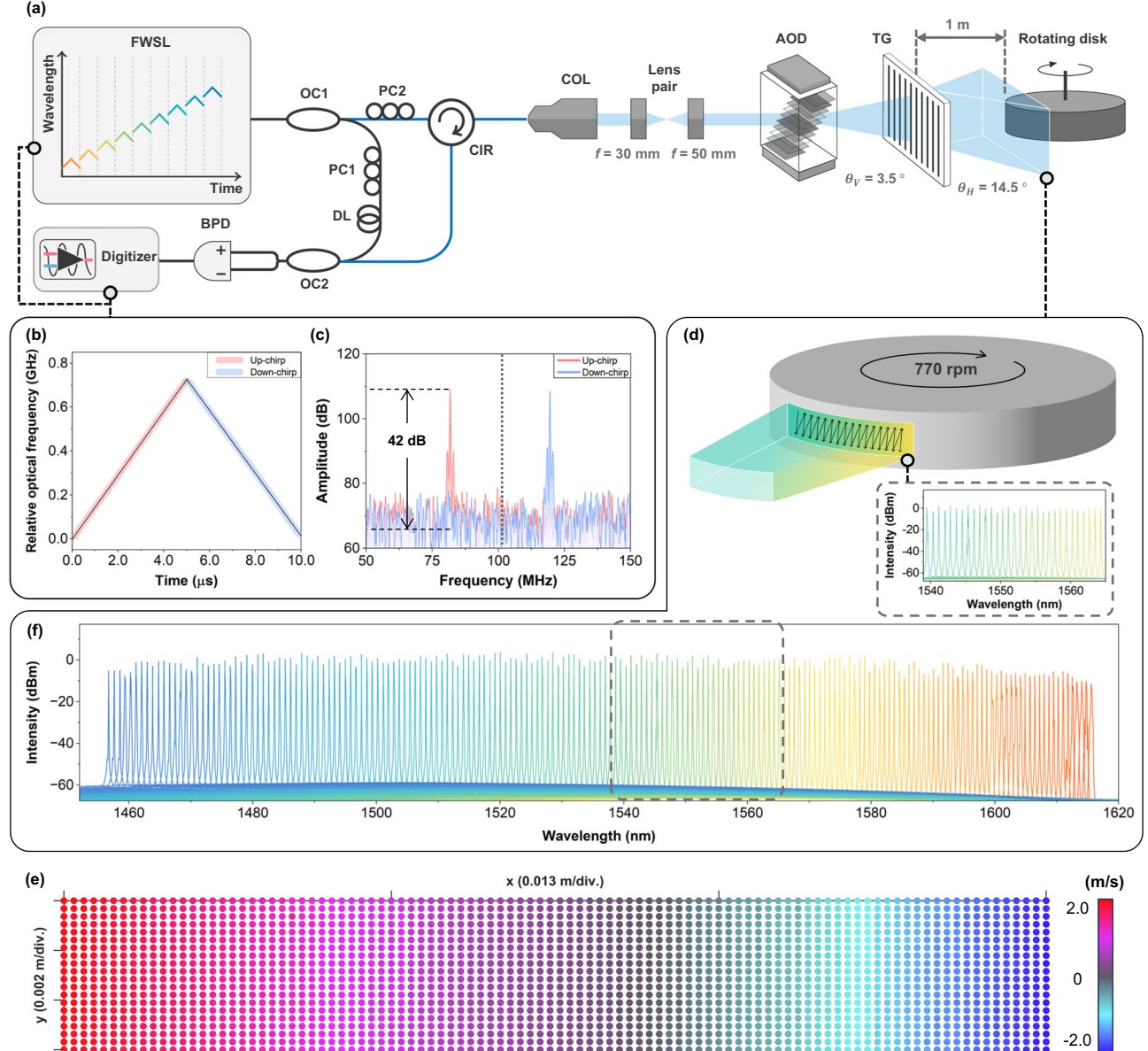

**Fig. 2 | Characterization of the FWSL and coherent LiDAR system.**
**a** Experimental setup of coherent LiDAR for system characterization. The solid blue lines represent the sample arms. The light in the sample arm propagates to free space via the collimator and passes a lens pair ($f$ = 30 and 50 mm). A solid-state beam scanner consisting of the AOD and TG is aligned for vertical and horizontal beam steering, respectively. **b** Wavelength modulation results obtained during a single flutter-wavelength modulation. The colored background and black lines in the center depict the relative optical frequency variation and the linear polynomial fitted function of the optical frequency variation during the linear wavelength sweep, respectively. The R-squared values of the up- and down-chirp modulations

are 0.99997 and 0.99987, respectively. **c** Fast Fourier transforms the result of the interference signal by a single EOPM modulation. An interference signal with a 42 dB signal-to-noise ratio is obtained using a retroreflective target placed at 20 m. The acoustic frequency shift is indicated by a black dotted line corresponding to $2f_a$. **d** Target description. A 2D raster scan is performed on the central area of the rotating disk, where a linear change in velocity occurs, including the zero-velocity part. **e** Spectral bandwidth of the FWSL measured by a discrete wavelength sweep of the WS. **f** Ten frames of stacked 3D velocity images (100 × 20 pixels) of the scanned area.

center wavelength, respectively. The region of interest (ROI) was measured 100 times to verify the accuracy and precision of the proposed LiDAR system.

As a result, we successfully obtained a clear 3D velocity image, as shown in Fig. 2f. To provide more in-depth verification, we analyzed the velocity distribution along a single spatio-spectral axis of the 3D velocity image. In Fig. 3a, the linear velocity changed from positive to negative, as expected from the alignment of the rotating disk. This velocity distribution closely followed the theoretical velocity indicated by the black dashed-dotted line (Fig. 3a), which was calculated relative to the propagating beam direction within the horizontal FOV[30]. For

comprehensive calculations, please refer to the Supplementary Note 10.

In Fig. 3b, we present error values, all within magnitudes not exceeding 0.08 m/s across the spatio-spectral axis. These errors may have originated from unintended vibrations of the disk and its inherent ellipticity. Nevertheless, it is evident that the measured velocities remained reliable within a reasonable error rate of ~5%, except in the zero-velocity region.

To further demonstrate the velocity measurement precision of our system across the spatio-spectral axis, we provided the standard deviation (SD) of 100 measured values at each lateral point, as shown

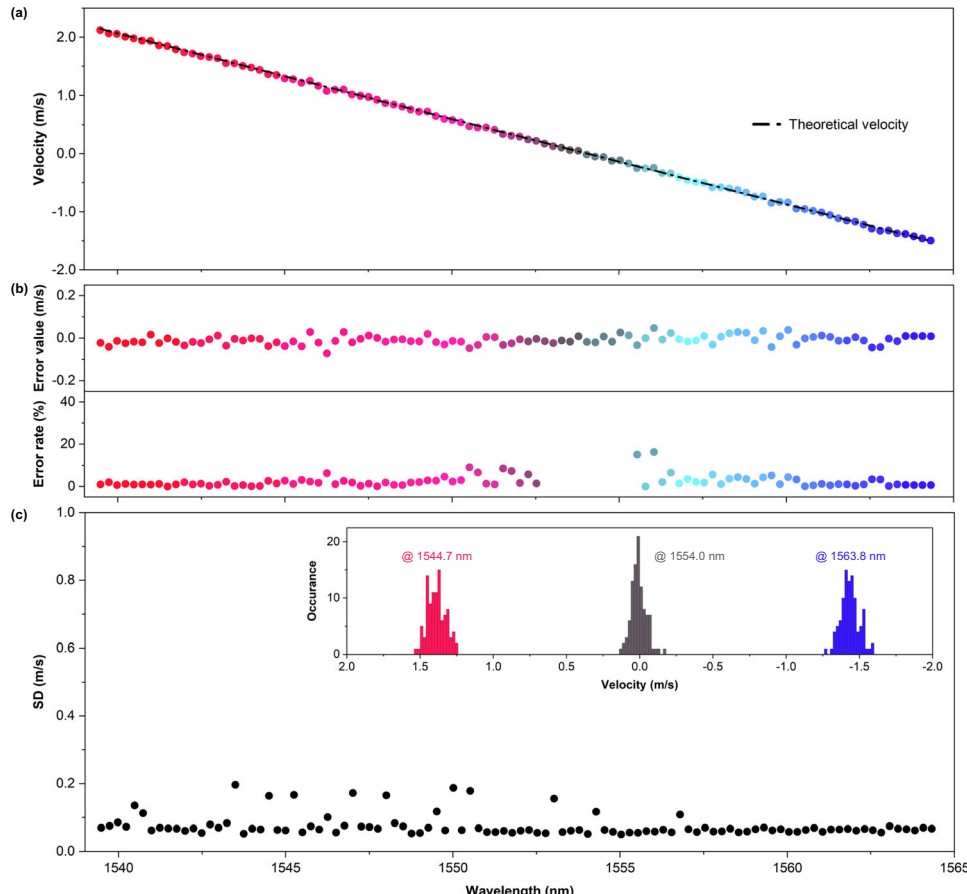

**Fig. 3 | Velocity measurement accuracy and precision. a** Velocity distribution along a single spatio-spectral axis of the 3D velocity image (Fig. 2f). The theoretical velocity for the field of view (FOV) is calculated based on the propagating beam direction to the disk, which is aligned at 1 m distance from the TG at a normal incidence of the beam. **b** Error value and error rate of the velocity measurement result in **a**. **c** Standard deviation (SD) of 100 velocity measurements along the same spatio-spectral axis as that of **a**. The right inset displays histograms depicting velocity measurements at wavelengths of 1544.7 nm, 1554.0 nm, and 1563.8 nm, corresponding to the positive, zero, and negative velocities, respectively.

in Fig. 3c. Notably, the SD was recorded consistently within the 0.1 m/s range throughout the spectral scanning from 1549 nm to 1564 nm. A few spatio-spectral points exhibited relatively high SDs but did not exceed 0.2 m/s. Considering the native velocity resolution determined by the FWSL, the obtained velocity results were within a reasonably accurate range.

We selected three representative points for each of the positive-, zero-, and negative-velocity parts among the different spatio-spectral points for further investigation of precision. The histograms of the measured velocity during 100 measurements at three selected wavelengths of 1544.7 nm, 1554.0 nm, and 1563.8 nm are provided in the inset of Fig. 3c. A clear tendency of a normal distribution was observed at all points. The mean error values of the estimated normal distribution at each point were recorded as 0.02 m/s, 0.03 m/s, and 0.003 m/s, respectively, indicating the accuracy of our experimental velocity measurement. The corresponding SDs were 0.063 m/s, 0.052 m/s, and 0.062 m/s, respectively, indicating that the actual velocity measurement was more precise than the theoretical velocity resolution. The proposed system successfully performed precise velocity measurements during spatio-spectral scanning, demonstrating the unique strength of the proposed FWSL. This achievement is notable among the other spatio-spectral coherent LiDARs based on a wavelength-swept laser reported to date.

In addition to the investigation of velocity measurement accuracy and precision described previously, we have also conducted an evaluation of distance measurement accuracy and precision. Employing the same experimental setup illustrated in Fig. 2a, we executed one-dimensional (1D) axial coherent distance measurements on a static target. The distance of a retroreflective tape positioned at distances ranging from 1 to 25 m relative to the TG is measured 1000 times. As illustrated in Supplementary Fig. 14, axial coherent ranging effectively exhibits distance measurements up to 25 meters, showing a linear increase of average measured distance. Remarkably, the SDs at all distances are recorded within a range of 0.04 m, which indicates that the actual distance measurements were more precise than the axial resolution determined by FWSL.

## 4D coherent ranging results

We conducted comprehensive 4D coherent measurements using the proposed LiDAR system under various scenarios. These scenarios, denoted as Scenes A, B, and C in Fig. 4a, were designed to demonstrate short-range (<7 m) and long-range (>20 m) 3D distance imaging and real-time 4D distance and velocity imaging, respectively. Photographs of the actual scenarios are provided in Supplementary Fig. 17.

Scene A was designed for high-resolution imaging within a room-scale distance range. We placed the objects—a model bus, a police car, and a tunneled bridge—at distances of 4.8, 5.4, and 6 m, respectively, against a background screen at 7.3 m. The 3D distance image was obtained using an FOV of 11.5° × 3.5° (H × V) and a fast axis scan rate of 0.5 kHz with dimensions of 600 × 190 pixels. Various miniatures in real-world settings were successfully imaged, as shown in Fig. 4b. Objects with distance differences of several tens of centimeters were

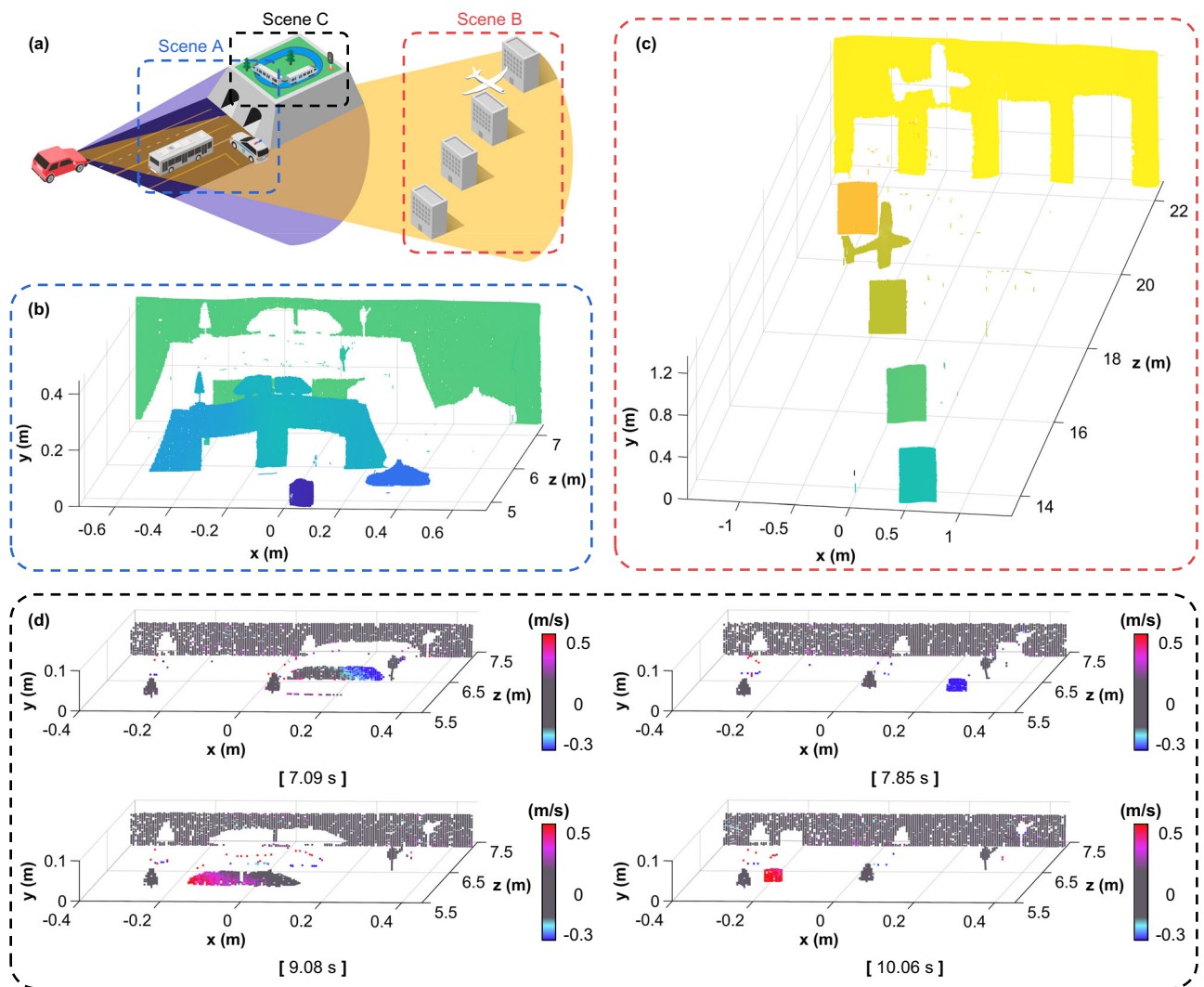

**Fig. 4 | 4D coherent ranging results. a** Schematic of different coherent measurement scenarios. Scenes A and B are designed for 3D distance measurements in the short-range (<7 m) and long-range (>20 m), respectively. Scene C is designed for real-time 3D distance and velocity measurements. **b–d** Imaging results for each scenario in the 3D point-cloud. **b** 3D distance image of Scene A sized 600 × 190 pixels at a frame rate of 2.6 Hz. **c** 3D distance image of Scene B sized 600 × 190 pixels at a frame rate of 2.6 Hz. **d** 4D distance and velocity images of Scene C sized 200 × 45 pixels at a frame rate of 10 Hz. Four individual frames are captured from a full-time video of 11.6 s, which is provided in Supplementary Video 1.

obviously distinguished within a relatively short distance range of 7 m or less. Even small objects, such as toy trees and traffic lights, were clearly imaged in their detailed shapes, demonstrating the excellent imaging resolution of the proposed system. We have provided a detailed post-processing procedure for the obtained images in the Methods section.

Scene B, which spanned over 20 m, was designed to demonstrate the capability of the proposed system for long-distance imaging. We set up a wall at 22.5 m as the background and positioned retroreflective boxes at 2-m intervals. A model airplane was positioned on the second box from the back at 18.5 m. A long-range 3D distance image consisting of 600 × 190 pixels was obtained with an FOV of 7.7° × 3.5° (H × V) and a fast axis scan rate of 0.5 kHz. As shown in Fig. 4c, it is confirmed that the coherent measurement over a substantial distance was successfully performed. Objects over a few tens of meters were measured, in addition to the model airplane, which was clearly distinct in its complex shape in the image. The obtained 3D distance images indicate that the proposed LiDAR system is capable of both short- and long-distance imaging while maintaining a high-resolution for desirable imaging quality in each scenario. Scenes A and B were designed differently to demonstrate high-resolution and long-distance imaging, respectively.

However, our system is capable of simultaneously imaging from short to long distances.

Scene C, which contained both stationary and moving objects, was designed to demonstrate real-time 4D imaging using the proposed coherent LiDAR system. With respect to the same target configuration in Scene A, we measured a train moving at a constant speed on a rail with a size of 0.47 × 0.69 m (W × L) on a tunneled bridge. Two trees were positioned alongside the rail—one at the start and the other in the middle—and a traffic light was positioned at the end. The 4D distance and velocity images of the ROI sized at 200 × 45 pixels were obtained continuously with an FOV of 7.7° × 1.24° (H × V) and a fast axis scan rate of 2 kHz. We achieved a frame rate of 10 Hz by increasing the fast axis scan rate to 2 kHz and adjusting the pixel size to suit the reduced ROI. A full-time video of 11.6 s for the obtained real-time 4D images synchronized with the actual picture in Scene C during measurement is provided in Supplementary Video 1. We captured images from the video, as shown in Fig. 4d, to confirm that the 3D distances and velocities of both the stationary and moving objects were clearly measured simultaneously. As shown in both the video and figure, it is obvious that distinct 4D imaging of the objects was performed without any interruption, indicating that the

proposed LiDAR system provided sufficient distance measurement resolution and lateral resolution.

## Discussion

In this study, we introduce an FWSL designed for spatio-spectral coherent LiDAR and a solid-state LiDAR system, showcasing comprehensive solid-state 4D coherent measurements. The FWSL is a laser source that enables independent bidirectional wavelength modulation of 5–6 pm throughout the continuous wavelength sweep for its wide bandwidth of 160 nm, centered at 1535 nm. The proposed system successfully performed simultaneous distance and velocity measurements using decoupled axial coherent ranging and lateral spatio-spectral scanning. In addition to the spectral scanning facilitated by the FWSL, we achieved solid-state 2D scanning using another non-mechanical scanner, AOD. This combination enabled effective real-time (10 Hz) 4D imaging relative to the long-distance 3D imaging of miniatures. The results demonstrate the robustness and efficiency of our LiDAR system, indicating its potential for a wide range of remote sensing applications.

Our system exhibits unique characteristics compared to previously reported spatio-spectral coherent LiDAR systems, which we outline as follows:

1) Flutter-wavelength modulation is a standout feature of our system that remains independent throughout the spectral bandwidth, thereby enabling 4D coherent ranging over extensive distances. Although the promise of spatio-spectral coherent LiDAR is evident owing to its inherent strengths, previously reported systems have limitations in velocity measurements along with relatively short measurable distances. We marked a significant advancement with the FWSL, which addressed both limitations by instantly and sequentially sampling the interference signals of flutter-wavelength modulation. In previous studies[21,24], only unidirectional wavelength sweeps were achieved, limiting the system to distance measurements. In addition, the interference signal across the entire wavelength bandwidth was measured simultaneously, placing a considerable burden on the electrical bandwidth, such as the response bandwidth of the photodetector, and sampling rate of the digitizer, considering the Nyquist theory. Consequently, this imposed a strong limit on the maximum measurable distance. In contrast, the FWSL supports independent wavelength modulation, particularly exhibiting a fluttering shape, during continuous wavelength sweeps within the spectral bandwidth, which enables velocity measurement. Instant and sequential sampling of the interference signal of the flutter-wavelength modulation substantially extended the theoretical maximum measurable distance to 264 m. We experimentally demonstrated velocity measurements during spatio-spectral scanning and coherent measurements over 20 m using a BPD of 400 MHz and a digitizer of 500 MS/s. A detailed comparison between our method and conventional approaches is provided in Supplementary Fig. 1 and Supplementary Table 1.

2) Flexible operation of the spectral bandwidth is another strength of the proposed system, which was achieved by decoupled axial coherent ranging and lateral spatio-spectral scanning. In previous studies, deliberate windowing upon the spectral bandwidth was required to arrange the axial distance and lateral position information from the interference signal across the entire spectral bandwidth. In contrast, our system provides the entire bandwidth for spatio-spectral mapping, regardless of complex signal post-processing accompanied by a direct trade-off between axial and lateral resolutions. The spectral bandwidth for a wavelength sweep can be adjusted according to the spatial ROI, which indicates that the bandwidth can be used as required for a wide FOV regardless of the measurable distance. Thus, spectral scanning conditions can be flexibly set by considering the desired imaging resolution and frame rate.

3) A flawless solid-state coherent LiDAR was achieved by introducing another solid-state beam-scanning mechanism in addition to

spatio-spectral mapping. This shows a meaningful approach not only for our system but also for other coherent LiDAR systems in terms of solid-state beam scanning. Active scanning in the vertical direction was performed using a high-speed, high-resolution non-mechanical beam scanner AOD. It was able to ignore the effect of internal interference caused by interferometer mating parts or surface reflections in the optical system owing to the acoustic frequency shift. We successfully achieved real-time, high-resolution 4D imaging at a frame rate of 10 Hz with a fast axis scan rate of 2 kHz, enabled by the AOD. However, there is still room for improvement to attain a higher frame rate using AOD, which offers high-speed scanning up to several MHz, as long as a higher acquisition rate is assured. In addition, the AOD is controlled electrically in the triggered state with the EOPM and WS, which are also controlled electrically. The electric-based synchronized system, which especially operates in a solid-state, ensures robust data acquisition regardless of external turbulence or optical jitter.

Despite its unique advantages, there remain several challenges to be addressed in our system. The primary limitation was the native axial resolution determined by the FWSL. The theoretical axial resolution of 0.21 m, which is reasonable for long-range measurements, should be improved considering the specifications desired for real-world applications. Since the present axial resolution is determined by the flutter-wavelength modulation bandwidth, we suggest that the axial resolution can be improved by increasing the modulation bandwidth through careful optimization of the FWSL architecture. The FWSL, which is based on an external cavity structure with a Littrow configuration[33], exhibits a decrease in the modulation bandwidth as the cavity length increases[34,35], assuming the same EOPM operation. We have provided a more detailed explanation of the architecture and principles of the FWSL in the Supplementary Note 2–5. Therefore, optimization of the cavity length of the FWSL can enable a wider modulation bandwidth. In addition, the modulation bandwidth can be increased by applying a higher amplitude modulation signal to the EOPM. Throughout the experiments, the modulation signal for the EOPM remained fixed at optimal conditions, as determined based on the flutter-wavelength characteristics illustrated in Supplementary Fig. 2 and 3. Currently, it is observed that the response of the used voltage amplifier (VA) appears insufficient under high amplitude or high-frequency conditions. By employing a high-performance VA, the modulation bandwidth can be expanded directly by applying a higher voltage to the EOPM. Based on these strategies, we suggest that the axial resolution is due to the currently fabricated FWSL and does not arise from a fundamental flaw in the FWSL architecture. However, the axial resolution does not necessarily define the distance measurement resolution of our LiDAR system. Regarding LiDAR is a technology for topography over the meter scale, the practical distance and velocity resolutions for the measurements should be considered as the precision of the overall system. As demonstrated above, our system supports distance resolution under 0.04 m and velocity resolution around 0.06 m/s. Supplementary Video 1, which shows a distinct 4D image of the objects without any interruption, experimentally proves that the actual measurements were more precise than the axial resolution. In addition, we expect that there is also room for improvement in the distance and velocity resolutions of measurements through the increase in the number of samples using common zero-padding[24].

Another limitation is the FOV along the acousto-optic axis. The scan angle of the AOD, which corresponds to the vertical FOV, was not sufficiently wide compared with the desirable vertical FOV. At present, most non-mechanical scanners, including acousto-optic devices, lack the wide FOV necessary to replace traditional mechanical scanners. Therefore, ongoing investigations are pursued to implement wider FOVs. However, this limitation can be practically mitigated by incorporating relay optics into the optical system[36] or introducing an additional scan lens. Furthermore, considering the unique operation of the FWSL due to its wide spectral bandwidth, the implementation of

other solid-state scanners[37], such as 2D dispersers[22], can be an accessible alternative. Additionally, in consideration of the future development of coherent LiDAR, we expect that the proposed FWSL can branch into various future studies. As faster and more compact light sources and LiDAR systems are preferred, an approach to parallel LiDAR based on simultaneous multi-wavelength generation[5,23], or miniaturization of the FWSL through integration into PIC-based lasers[38], will be meaningful for further studies.

In conclusion, we proposed a unique wavelength-swept laser, FWSL, and a comprehensive 4D solid-state coherent LiDAR system based on the FWSL. Simultaneous coherent measurements of distance and velocity, according to horizontal spatio-spectral mapping, were achieved using independent flutter-wavelength modulation during the wavelength sweep by the FWSL. Furthermore, we incorporated vertical solid-state scanning via AOD to enable high-resolution 2D solid-state beam scanning. As 4D coherent measurements in various scenarios were successfully performed, high-resolution 3D distance images were obtained from room-scale (within 7 m) to long-range (over 20 m), and 4D distance and velocity images were obtained in real time. Furthermore, we demonstrated the robustness of the proposed coherent LiDAR system even under challenging conditions by implementing real-time 4D imaging in a simulated fog environment (Supplementary Video 2). The proposed system stands as a robust, efficient, and straightforward spatio-spectral coherent LiDAR system with promising implications for the future of LiDAR technology. This advancement represents a significant step towards the realization of advanced coherent LiDAR applications.

## Methods

### Electrical system operation
Electrical control of the overall system was based on an AFG (Tektronix, Beaverton, OR, USA) and a data acquisition (DAQ) board synchronized with the LabVIEW program. First, the AFG generated a triangular waveform with a 10 $V_{pp}$ amplitude and a 100 kHz rate. The generated signal was applied to an EOPM (Thorlabs, Newton, NJ, USA) after the voltage was amplified 20 times through a VA (Thorlabs, Newton, NJ, USA). Subsequently, the control signal of the AOD (Photon Lines, Banbury, UK) programmed in LabVIEW was applied to a variable-frequency driver (Photon Lines, Banbury, UK) through the DAQ board. An acoustic frequency corresponding to the voltage from the DAQ board was applied to the AOD. Here, the AOD was driven by a ramp waveform, and its amplitude (vertical FOV) and speed (fast axis scan rate) were adjusted according to the target scenarios. Finally, a WS (Thorlabs, Newton, NJ, USA) was driven by a step waveform generated by the LabVIEW program and applied directly through the DAQ board. Similarly, the amplitude (horizontal FOV) and speed (frame rate) were set depending on the target scenarios. The AOD was triggered to EOPM synchronously with the transistor-transistor logic (TTL) signal of the AFG, and WS was triggered to AOD by the TTL signal generated by the LabVIEW program.

### Characterization of FWSL
The relative optical frequency during wavelength modulation by the EOPM was measured to characterize the FWSL. A 4 × 4 Mach–Zehnder interferometer was used for the measurements, with the selected wavelength at the center. The phase variation during the single-wavelength modulation was extracted and converted into an optical frequency[39,40] (Fig. 2b). The detailed procedure is given in Supplementary Note 7. The spectral bandwidth of the FWSL was verified by measuring the change in wavelength by WS using an optical spectrum analyzer (OSA; Anritsu, Atsugi, JP) with a 0.03 nm resolution. Different voltages were applied to the WS at certain intervals, which selected the wavelength discretely. The output of the FWSL was directly measured using the OSA, and the results of 191 measurements are shown in

Fig. 2d. The optical power of the FWSL at 1535 nm was measured as 20 mW. During the overall operation of the FWSL, a constant typical operating current and temperature, which may primarily affect optical power, were provided to the SOA.

### Design of optical system
A fiber collimator (Thorlabs, Newton, NJ, USA) was first placed in a free-space optical system. The collimated beam was passed through an achromatic lens pair (Thorlabs, Newton, NJ, USA) with focal lengths of 30 mm and 50 mm. The spacing between the two lenses was adjusted to focus the beam at the desired distance for each experiment. The AOD was placed after the lens pair in line and aligned for vertical beam deflection. A TG (Edmund Optics, Barrington, NJ, USA) was located after the AOD to cover the diffracted light within the TG area for horizontal beam deflection. The TG with a groove density of 1000 grooves/mm was aligned at a 50° angle with the propagating light to satisfy the Littrow angle condition. The FOV in the vertical and horizontal directions under these conditions were observed as 3.5° and 14.5°, respectively. The final output power at the end of the TG was ~5 mW. The focused beam at 1 m was measured ~2 mm of beam spatial profile.

### Data acquisition and processing
The optical interference signal was measured using a BPD (Thorlabs, Newton, NJ, USA) with a 400 MHz bandwidth. The measured signal was collected on channel 1 of the digitizer (Alazartech, Quebec, CA) at a sampling rate of 500 MS/s, and a reference signal from the variable-frequency driver was collected simultaneously on channel 2. Here, the reference signal had the same frequency as the acoustic frequency applied to the AOD, but a power of 0 dBm. The data for a single-wavelength modulation from channel 1 and the corresponding data from channel 2 were digitized in 4960 samples. The signal acquired for each channel was divided into up- and down-chirps, and zero-padding was performed for up to 4992 samples in each chirp. After applying the FFT to each chirp, the detected beat frequency from channel 1 and the acoustic frequency from channel 2 was obtained by extracting the peak frequency at each chirp (Fig. 2c). The true beat and Doppler frequencies were obtained by compensating the shifted acoustic frequency for the measured beat frequency according to Eqs. 3 and 4. For image processing, a 2D median filter was applied to the 3D distance images obtained through a previous data-processing procedure. Subsequently, the built-in statistical outlier removal filter of the CloudCompare program was applied and rasterized by interpolation. The final 3D point-cloud data were plotted using MATLAB.

### Target configuration for 4D coherent ranging
The design of Scene A for room-scale 3D distance imaging is illustrated in Supplementary Fig. 4a. The first bus, police car, tunneled bridge, and last white screen were located at 4.8 m, 5.4 m, 6 m, and 7.3 m from the TG, respectively. A reflective spray (ALBEDO 100, Sweden) was applied to the surface to enhance the reflectivity of the targets. The design of Scene B for long-range 3D distance imaging is depicted in Supplementary Fig. 4b. The first box is on the far right; the second, third, and fourth boxes; and the last wall was located at 13.5 m, 16 m, 18.5 m, 21 m, and 22.5 m from the TG, respectively. A reflective tape (3 M, Saint Paul, MN, USA) was applied to the boxes to enhance reflectivity, and a reflective spray was sprayed onto the surfaces of the model airplane and wall. The design of Scene C for simultaneous 4D distance and velocity imaging is shown in Supplementary Fig. 4c. The upper part of the tunneled bridge in Scene A was set as the ROI, and the stationary models (two model trees and a traffic light) were located at intervals of 0.3 m from 6 m, in the order mentioned. Similarly, a reflective spray was applied.

## Data availability
The data that support the findings of this study are available from the corresponding author upon request.

## Code availability
The codes used throughout this study are available from the corresponding author upon request.

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

## Acknowledgements

The authors thank Professor Hwidon Lee for the discussion and valuable suggestions. This research was supported by the National Research Foundation of Korea (NRF), funded by the Ministry of Science and ICT, Korea (Grant no. NRF-2021R1A5A1032937), the Korea Evaluation Institute of Industrial Technology (KEIT) and Technology Innovation Program funded by the Ministry of Trade, Industry & Energy, Korea (Grant no. 1415181752, 20026554), and Hyundai Motor Company.

## Author contributions

D.J., H.J., and C.-S.K. contributed to the design of the system and experiments; D.J. and H.J. constructed the experimental setup; M.U.J. developed software programming for data acquisition; T.J., H.K., S.Y., and J.L. assisted in the preparation of the experimental setup; D.J., H.J., and M.U.J. obtained, analyzed, and processed the data. D.J. wrote the manuscript with input from all authors, and C.-S.K. reviewed it. C.-S.K. supervised the work.

## Competing interests
The authors declare no competing interests.
