## [Peer Review File · Nature Communications]

REVIEWER COMMENTS

Reviewer #1 (Remarks to the Author):

The authors have demonstrated a spatio-spectral 4D coherent LiDAR based on a flutter-wavelength-swept laser and demonstrated the detection results under different scenes, including both static and dynamic targets. Although the 4D imaging results are impressive, the sequential sampling process of the proposed spatial-spectral LiDAR essentially limits the acquisition rate, which is about 105 points/sec. The relatively low acquisition rate has long been the main bottleneck for coherent LiDARs compared with ToF LiDARs, and the proposed spatio-spectral LiDAR does not show obvious advantages to this problem. In addition, there inherently exists a trade-off between ranging resolution and acquisition rate for the proposed system. If the attempt is made to increase the detection points in the horizontal axis (fast-axis), the linear frequency-modulation period should be decreased and so does the frequency swept bandwidth, since the slope of the frequency sweep cannot be too large to avoid generating beating frequency exceeding the response bandwidth of the avalanched BPD. Thus, the ranging resolution must be sacrificed to maintain the maximum measurable distance.

Due to the drawbacks exist in the proposed LiDAR system, I don't think it can be published on Nature Communications.

Besides, some other questions needed be identified and addressed for further improvement:

1. Since flutter-wavelength-swept laser is one of the main important parts in this work, I suggest the authors add more details on it. The current schematic of the flutter-wavelength-swept laser configuration in Fig.1a is not well demonstrated, and the authors intentionally reduce the description of Littrow-type ECDL and does not clearly show its full structure in the schematic, which will bring unnecessary confusion to the readers.
2. The statement that "short measurable distance due to the wavelength sweep bandwidth" is not very accurate. In fact, larger slope of the linear frequency sweep can lead to higher ranging resolution and higher beating frequency. This is good when the response bandwidth of the detectors is not considered, but in practice the response bandwidth of highly sensitive detectors is usually limited (400 MHz for a typical avalanched BPD) and it acts as the main reason for limitation of the measurable distance. The authors should modify their statements.
3. 5 mW output optical power for a 6-7 m distance measurement is reasonable. Does the author use the same power level for 20 m distance measurement? The SNR of the beating frequency signals should be given.
4. The achromatic lens used in this work can only ensure collimation within a short range. Can it still ensure that no obvious caustics occur at 20m?

5. What's the modulation efficiency of the phase modulator that used in the FWSL? Why the proposed FWSL has good frequency-sweep modulation linearity? The unit of the amplitude of the triangular waveform in the supplementary material is also missing.

6. Why does the frequency-sweep modulation bandwidth have the relationship with the cavity length of ECDL? This content that could have become advantages should be discussed and highlighted in the main text.

7. 0.21m ranging resolution maybe reasonable to the long range detection, but how to solve the problem of poor range resolution of close range targets?

Reviewer #2 (Remarks to the Author):

D. Jeong and colleagues present a real time coherent distance and velocity measurement system based on the modulation of a semiconductor external cavity diode laser using a conventional grating pivot mechanism for coarse tuning and an intracavity phase modulator for fast fine tuning. Hence the beam scanning and the ranging modulation in spectral-spatial LiDAR using the flutter-wavelength-swept laser can be decoupled to improve performance. This paper solves one of the issues of earlier works by Okano et al. and achieves a 90 kHz pixel measurement rate, 10 Hz frame rate, 750 MHz flutter chirp bandwidth, corresponding to a 20 cm resolution, with a power of 5 mW power on the transmission grating aperture. The paper is in general well written and the figures are with some minor exceptions well crafted. Overall, I support publication in Nature Communications but would ask the authors to answer the following questions and comments:

1. Generally the authors should give more information on the FWSL, which is the core innovation of the paper. What are the free-spectral range of the laser, which should limit the optical linewidth? What limitations exist on the EO tuning range? I think there should be a trade-off between the fast-tuning range and the optical linewidth. Careful simulations of the relationship between laser linewidth showed us that a linewidth of less than 50 kHz is desirable for long range LiDAR beyond 150m. Can this be achieved while also achieving a cm-scale distance resolution with the current FWSL architecture? What is the architecture of the laser (pivot mechanism or simple Littrow grating? Is the coarse wavelength sweep mode hop free?

2. Why is the laser linewidth measurement with the self-heterodyne interferometer (see Supplementary Figure 1a) so noisy? What was the fiber delay length in the interferometer chosen for the measurement?

3. The Vpi modulation voltage requirement of the EOPM does increase with wavelength. How does the flutter modulation bandwidth with the EOPM change with respect to laser wavelength given the relative bandwidth of 10% of the laser sweep? It seems the authors have done all the measurements with a constant flutter modulation voltage. How large would the ranging error be?

4. What is the output power of the laser? Do the authors require an amplifier to achieve imaging without the reflective tape? What is the SNR of the beat notes on the 20 m non-reflective wall?

5. Is the metric for linearity the RMS deviation or the maximum error? Please also plot the deviation from linear fit in Figure 1b or in the supplement. It is impossible for the reader to verify the strong claim of 0.99997 linearity based on the presented data. Plotting the chirps as the relative wavenumber is quite unusual in the field of coherent LiDAR where usually a frequency axis is chosen.

6. What do the authors refer to with the measured linearity in line 269? Is it the linearity of velocity measurement? This should be a sine function not a linear function.

7. All practical imaging experiments (see Fig 4.) are performed using a horizontal FOV substantially smaller than the value of 11.1° quoted in line 220. Do the authors find a degradation of the imaging quality and resolution at the edges of the FOV?

8. Please reconsider your choice of color map. The yellow to brown colors used to depict distances are very hard to distinguish for the reader. Fig 2(f) is unreadable. The velocity color maps is better but positive and negative values around "0" are also hard to distinguish for the reader as they are both shades of grey and purple. Please add axis labels for Figure 2, Panels (f) and (g).

Reviewer #3 (Remarks to the Author):

This work aims to perform a solid-state coherent LiDAR. The authors propose a novel flutter-wavelength-swept laser that offers a simultaneous yet independent wavelength modulation of 5–6 pm for axial distance ranging and velocity measurement, and a wavelength sweep of 160 nm for horizontal beam scanning. However, there exists a severe concern that the experiment is limited and

lacks enough evidence for technique validation. The following concerns should be carefully addressed.

1. The introduction could benefit from a more detailed explanation of the performance and immaturity of silicon chip-based technology in relation to Coherent LiDAR. It would be helpful to clarify the limitations of this technology when compared to scanning-based Coherent LiDAR, allowing readers to better understand the research's contribution to the field.

2. To demonstrate the effectiveness of your method in comparison to conventional swept LiDAR, it would be useful to include additional quantitative results that showcase differences in distance and velocity accuracy. You should provide these quantitative comparative results compared with other LiDAR methods in the manuscript.

3. When considering the motion of the LiDAR in your method, it is important to address how data collection and any jitter during the process are managed. Elaborating on this aspect will provide greater insight into the system's functionality under various conditions.

4. While the article mentions the use of a spectral domain method for the LiDAR technique, it is necessary to discuss its performance in challenging environments such as rain, fog, and areas with pollution or smoke. Including further explanation and data-supported evidence for these scenarios will ensure a more comprehensive evaluation of the method's effectiveness across different conditions.

5. Although the images in the article are visually appealing, it would be beneficial to include a photograph of an actual experimental setup, showcasing each device and its placement. This addition will help readers gain a clearer understanding of the experiment's execution and enhance the overall presentation of the research.

Response to Reviewers' comments

Manuscript Number: NCOMMS-23-08645

Title: Spatio-spectral 4D Coherent Ranging Using a Flutter-wavelength-swept Laser

Authors: Dawoon Jeong, Hansol Jang, Min Uk Jung, Taeho Jeong, Hyunsoo Kim, Sanghyeok Yang, Janghyeon Lee, and Chang-Seok Kim

[Reviewer 1]

The authors have demonstrated a spatio-spectral 4D coherent LiDAR based on a flutter-wavelet length-swept laser and demonstrated the detection results under different scenes, including both static and dynamic targets. Although the 4D imaging results are impressive, the sequential sampling process of the proposed spatial-spectral LiDAR essentially limits the acquisition rate, which is about 105 points/sec. The relatively low acquisition rate has long been the main bottleneck for coherent LiDARs compared with ToF LiDARs, and the proposed spatio-spectral LiDAR does not show obvious advantages to this problem. In addition, there inherently exists a trade-off between ranging resolution and acquisition rate for the proposed system. If the attempt is made to increase the detection points in the horizontal axis (fast-axis), the linear frequency-modulation period should be decreased and so does the frequency swept bandwidth, since the slope of the frequency sweep cannot be too large to avoid generating beating frequency exceeding the response bandwidth of the avalanched BPD. Thus, the ranging resolution must be sacrificed to maintain the maximum measurable distance.

Due to the drawbacks exist in the proposed LiDAR system, I don't think it can be published on Nature Communications.

Besides, some other questions needed be identified and addressed for further improvement:

<Response>

We thank the reviewer for your meticulous review and valuable suggestions. While coherent light detection and ranging (LiDAR) is emerging as the next-generation technology to succeed in time-of-flight (TOF) LiDAR, it currently cannot entirely replace TOF LiDAR for several reasons. One of these reasons, as the reviewer mentioned, is that coherent LiDAR systems are currently slower. In other words, it is crucial to achieve a speed comparable to the acquisition rate of TOF LiDAR, which approaches a few megahertz.

To address this challenge, several studies have been proposed. Rogers et al.¹ introduced a flash LiDAR method that improved the speed of coherent LiDAR by using parallel detection for each pixel column. Riemensberger et al.² achieved an acquisition rate of 3 MHz by using a spectral microcomb generation and parallel detection, and Qian et al.³ recorded a 7.6 MHz acquisition rate using a high-speed broadband wavelength-swept laser and spectral deflection. Compared to the previous research, the speed of our system, based on a sequential operation and sampling process, may not be sufficient. However, we've successfully addressed several important challenges using our approach.

Content	Laser specification						
	Coherence length	Center Wavelength (nm)	Sweep bandwidth (THz)	Sampling point interval (nm/pt)	Axial resolution (cm)	Acquisition rate (MHz)	Fast axis scan rate (kHz)
Ours (Flutter-wavelength modulation)	> 1.18 km	1535	11.2 ^①	0.0056 ^②	21	0.1 ^④	2 ^③
Ours (Conventional wavelength sweep)	> 1.18 km	1535	11.2 ^①	0.44 ^②	0.27	0.4 ^④ (N=200)	2 ^③
Content	LiDAR system specification						
	Electric bandwidth		Maximum measurable distance (m)	Distance resolution (cm)	Velocity resolution (cm/s)	Frame rate (Hz)	
Photodetector (GHz)	Digitizer (GS/s)						
Ours (Flutter-wavelength modulation)	0.4	0.5	264	< 4	< 20	10 ^⑦ (200 ^⑥ x45 ^⑤ px)	
Ours (Conventional wavelength sweep)	0.4	0.5	1.7	< 4	< 20	44.4 ^⑦ (200 ^⑥ x45 ^⑤ px)	

Fig. 0.1. (a) Schematic of the simulation design (upper: flutter-wavelength modulation, lower: conventional wavelength sweep); **(b)** Laser and light detection and ranging (LiDAR) system specifications in the designed simulation.

We conducted some simulations to better understand the issues we've solved. We assumed a case where our flutter-wavelength-swept laser (FWSL) worked like typical wavelength-swept lasers. In this hypothetical situation, we kept major settings, such as the wavelength sweep bandwidth, fast axis scan rate, and imaging pixel size, the same as in our actual system. Moreover, the spectral axis was assumed to be the fast axis. We summarized the designs and results of the quantitative system for both our proposed system and this hypothetical system in Fig. 0.1.

Fig. 0.2. Simulation results of (a) maximum measurable distance along the acquisition rate; (b) Required electric bandwidth along the distance.

The simulation results for the maximum measurable distance, based on variations in the acquisition rate, are explored and illustrated in Fig. 0.2a. If the acquisition rate needs to be increased, the only option available is to reduce the maximum measurable distance, assuming that the flutter-modulation bandwidth and the electrical bandwidth remain constant. However, in the case where our proposed system operates under hypothetical conditions, the maximum measurable distance threshold would be approximately 160 times lower, indicating a significantly higher reduction rate.

Subsequently, we conducted a simulation on the required electrical bandwidth along the distance while keeping the other operating parameters constant. As shown in Fig. 0.2b, the required electrical bandwidth to cover the same extended distance in the proposed system is significantly reduced. If we assume a limit of 2.5 GHz for the electrical bandwidth, the maximum measurable distance for the hypothetical case is restricted to approximately 16 meters, which is less than ideal for remote sensing applications. This suggests that because there is a limit to increasing the electrical bandwidth, there is a limit to which the measurable distance cannot be extended any further, and in the hypothetical case, it is especially short.

These simulation results highlight that the proposed system utilizing the FWSL offers better resistance to electrical bandwidth. This allows for more freedom in optimizing the trade-off between modulation bandwidth and modulation speed while maintaining a higher maximum measurable distance threshold. Thanks to the flexible nature of the flutter-wavelength modulation and our sequential sampling approach, we've achieved substantial, measurable distances, reducing the electrical load through data-efficient operations along with the benefit of velocity imaging. Nevertheless, it is important to acknowledge that certain limitations persist, including insufficient axial resolution and acquisition rate. However, it is worth noting that all the simulation outcomes are derived from the parameters of our current system. We anticipate that we can address these weaknesses while preserving the strengths of our proposed system. We have provided a more detailed explanation in our response to the following comments to address them further.

We thank the reviewer for your thoughtful feedback and sincere suggestions. Based on your comments, we have made revisions to both the main manuscript and the supplementary information as follows.

[1] Rogers, C. *et al.* A universal 3D imaging sensor on a silicon photonics platform. *Nature* **590**, 256–261 (2021).

[2] Riemensberger, J. *et al.* Massively parallel coherent laser ranging using a soliton microcomb. *Nature* **581**, 164–170 (2020).

[3] Qian, R. *et al.* Video-rate high-precision time-frequency multiplexed 3D coherent ranging. *Nat Commun* **13**, (2022).

<Comment 1>

Since the flutter-wavelength-swept laser is one of the main important parts of this work, I suggest the authors add more details on it. The current schematic of the flutter-wavelength-swept laser configuration in Fig.1a is not well demonstrated, and the authors intentionally reduce the description of Littrow-type ECDL and does not clearly show its full structure in the schematic, which will bring unnecessary confusion to the readers.

<Response>

We thank the reviewer for your kind comments and sincere suggestions. We agree that our explanation of FWSL needs to be more detailed to ensure better comprehension for readers. Therefore, we have included an additional explanation regarding the structure and principles of FWSL, which is provided below:

Location: *Supplementary Information*

“FWSL structure and principles

The detailed structure of the FWSL is depicted in Supplementary Fig. 2. The FWSL adopts a Littrow configuration with a tunable reflective filter structure based on a wavelength selector (WS), which consists of a mirror and a reflective holographic diffraction grating (1050 grooves/mm). The broadband light generated from the gain is collimated by the collimator and passed through the electro-optic phase modulator (EOPM), which is for flutter-wavelength modulation. Then, an output wavelength is selected via the WS by adjusting the angle of light incident on the diffraction grating. The reflected light of the selected wavelength is lased within the cavity from the gain to the WS and is emitted through a fiber-coupled output. During the overall operation, the output wavelength sweep via WS and the flutter-wavelength modulation via EOPM are synchronized using an electrical control signal from an arbitrary function generator.

Supplementary Fig. 1. Detailed structure of FWSL. AFG and WS denote arbitrary function generator and wavelength selector, respectively.

The output wavelength of the FWSL is determined by a combination of the cavity spectrum of the gain chip, the longitudinal mode spectrum of the external cavity, and the reflection spectrum of the WS. The internal cavity gain of the gain chip can be described as follows^{3,4}:

$$G_{IC} = \frac{G}{(1 + G\sqrt{R_1 R_2})^2 + 4G\sqrt{R_1 R_2} \sin^2 \theta} \quad (1)$$

where G denotes the single pass gain, R_1 , R_2 indicate the reflectance of both facets of the gain chip, and θ refers to the phase difference that can be expressed as $\theta = 4\pi L_{IC}/\lambda$, where L_{IC} and λ denote the internal cavity length and wavelength⁴, respectively. The external cavity gain is expressed as⁵

$$G_{EC} = \frac{T_2}{[1 - G\sqrt{R_1 T_2 R_3 \sqrt{R_4}}]^2 + 4G\sqrt{R_1 T_2 R_3 \sqrt{R_4}} \sin^2 \delta} \quad (2)$$

where R_3 and R_4 indicate the reflectance of the angle scanner and diffraction grating, respectively, and δ denotes the phase difference, which has a relation of $\theta = 4\pi L_{EC}/\lambda$ with external cavity length, L_{EC} . The reflectance of a diffraction grating can be expressed as⁴

$$R_4 = R_g \left(\frac{\sin(N\phi/2)}{N \sin(\phi/2)} \right)^2 \quad (3)$$

where R_g denotes the peak reflectance of the grating, N corresponds to the illuminated groove number of the diffraction grating, and ϕ symbolizes the phase difference, which can be calculated by $\phi = 2m\pi\lambda_0/\lambda$, where λ_0 indicates the feedback wavelength of the grating. The

gain spectrum of the FWSL is determined by the product of the above factors expressed as follows⁴:

$$G_{\text{net}} = R_4 G_{IC} G_{EC} \quad (4)$$

For simplicity, the losses of the collimator and the EOPM are ignored.

Supplementary Fig. 3 shows the structural configuration of the cavity mode of the FWSL. The cavity configuration parameters for the simulation are listed in Supplementary Table 2. Based on these simulations, three identical lasers were fabricated, which operated well as intended, demonstrating the designed features of the FWSL.

Supplementary Fig. 2. Simulation results of the cavity mode structure of the FWSL.

Supplementary Table 1. Parameters and values for the cavity spectrum simulation.

Parameter	L_{IC} (mm)	L_{EC} (mm)	G	R_1	R_2 (T_2)	R_3	R_g	λ_g (nm)	N
Value	4.65	50.1	1	0.1	0.005 (0.995)	1	1	1550	10000

Principle of wavelength change

Previously, it was confirmed that the output wavelength of the FWSL is determined by a complex spectrum of the internal and external cavities, and each cavity gain is derived from the cavity length. The external cavity, which exhibits a relatively high gain, primarily influences the wavelength selection. In other words, the output wavelength can be changed by adjusting the length of the external cavity in the FWSL⁵. In addition, the output wavelength can be changed by varying the angle of the light incident on the diffraction grating³.

Supplementary Fig. 3. (a) Simulation results of G_{EC} according to L_{EC} change; (b) Resulting peak wavelength change.

The FWSL utilizes both techniques for wavelength changes. First, flutter-wavelength modulation was achieved by controlling the external cavity length through EOPM at each fixed angle of the incident light on the diffraction grating. Supplementary Fig. 4 illustrates the simulation results of the different gain spectrum when the external cavity length was changed. The remaining parameters, except L_{EC} used the same values as those in Supplementary Table 2.

Next, a wavelength sweep using the WS was implemented by changing the angle of light incident on the diffraction grating. In the Littrow configuration, the first-order feedback wavelength of the diffraction grating can be expressed as³

$$\lambda_g = 2d \sin \theta \quad (5)$$

where d and θ denote the grating period and the angle of light incident on the diffraction grating, respectively. Based on the gain spectrum of the external cavity (Eq. 2) along with the effect of the diffraction grating, only external cavity modes of a certain order are selected and survive, depending on the angle of light incident on the diffraction grating. Supplementary Fig. 5 presents the simulation results of the different gain spectrum when λ_g changes. For the simulation, the remaining parameters, except for λ_g , had the same values as those listed in Supplementary Table 2.

Supplementary Fig. 4. (a) Simulation results of G_{EC} based on the change in λ_g ; (b) Resulting peak wavelength change.

[3] Haiping, G., Chenhao, W., Fei, W., Lijing, Z. & Chengwen, X. Study on the dynamic mode stability of grating-feedback external cavity diode lasers. *Laser Phys* **26**, 045002 (2016).

[4] Gong, H., Liu, Z., Zhou, Y. & Zhang, W. Extending the mode-hop-free tuning range of an external-cavity diode laser by synchronous tuning with mode matching. *Appl Opt* **53**, 7878 (2014).

[5] Levin, L. Mode-hop-free electro-optically tuned diode laser. *Opt Lett* **27**, 237 (2002).

<Comment 2>

The statement that “short measurable distance due to the wavelength sweep bandwidth” is not very accurate. In fact, a larger slope of the linear frequency sweep can lead to higher ranging resolution and higher beating frequency. This is good when the response bandwidth of the detectors is not considered, but in practice the response bandwidth of highly sensitive detectors is usually limited (400 MHz for a typical avalanche BPD) and it acts as the main reason for limitation of the measurable distance. The authors should modify their statements.

<Response>

We appreciate the thoughtful comment. We admit that our original statement was inappropriate, not considering the relationship between various factors sufficient. The problem of a short measurable distance in spatio-spectral coherent LiDAR does not directly arise by the bandwidth of the wavelength sweep itself but rather by how the interference signal across the entire spectral bandwidth is sampled. To clarify this, we have revised our statement to provide a more accurate and specific explanation, as follows:

Location: page 5, line 85-96

“2) short measurable distance owing to the sampling process. In addition to the coherence length of the laser, the maximum measurable distance is calculated as $d_{max} = f_s c T / 4B$, where f_s , c , T , and B denote the sampling rate of the digitizer, speed of light, sweep period, and sweep

bandwidth²¹, respectively. Moreover, it is assumed that the beat frequency of the entire wavelength bandwidth does not exceed the response bandwidth of the avalanched balanced photodetector (BPD). Since the complete distance data were collected and sampled along a single lateral axis simultaneously, the measurable distance was significantly limited due to the finite electrical bandwidth compared to the provided wavelength bandwidth. In other words, there is a significant trade-off between the measurable distance, the sweep rate, and the wavelength bandwidth. Okano et al. extended the measurable distance from 0.25 to 12 meters by reducing the sweep rate and bandwidth to 300 Hz and 19 nm, respectively. However, this approach is less desirable because it results in the loss of the advantages associated with spatio-spectral mapping and real-time ranging.”

[21] Okano, M. & Chong, C. Swept Source Lidar: simultaneous FMCW ranging and nonmechanical beam steering with a wideband swept source. *Opt Express* **28**, 23898 (2020).

Location: page 6, line 117-120

“While the wavelength sweep is in progress, the flutter-wavelength modulation segment is instantly and sequentially sampled. This approach effectively resolves the measurable distance limitations that existed in the previous sampling method.”

Location: page 20, line 389-392

“In addition, the interference signal across the entire wavelength bandwidth was measured simultaneously; there was a considerable burden on electrical bandwidth, the response bandwidth of the photodetector, and the sampling rate of the digitizer, considering the Nyquist theory. Consequently, this imposed a strong limit on the maximum measurable distance.”

<Comment 3>

5 mW output optical power for a 6-7 m distance measurement is reasonable. Does the author use the same power level for 20 m distance measurement? The SNR of the beating frequency signals should be given.

<Response>

We appreciate your clarification and suggestion. Throughout our experiments, we consistently maintained an output power of approximately 5 mW. It’s noteworthy that this optical power may not be sufficient for measurements over 20 meters. To address the insufficient optical power, we enhanced the reflectivity of the targets by applying reflective sprays (ALBEDO 100, Sweden) and reflective tape (3M, Saint Paul, MN, USA). We acknowledge that our initial manuscript lacked clarity in explaining the target configuration, which may have caused confusion. To prevent further confusion, we have included a detailed description of the target configuration in the manuscript, which is presented below.

Location: page 25, line 528-539

“Target configuration for 4D coherent ranging

The design of Scene A for room-scale 3D distance imaging is illustrated in Supplementary Fig. 4a. The first bus, police car, tunneled bridge, and last white screen are located at 4.8 m, 5.4 m, 6 m, and 7.3 m from the TG, respectively. A reflective spray (ALBEDO 100, Sweden) was applied to the surface to enhance the reflectivity of the targets. The design of Scene B for long-range 3D distance imaging is depicted in Supplementary Fig. 4b. The first box on the far right, second, third, fourth boxes, and last wall are located at 13.5 m, 16 m, 18.5 m, 21 m, and 22.5 m from the TG, respectively. Reflective tape (3M, Saint Paul, MN, USA) was applied to the boxes to enhance reflectivity, and a reflective spray was sprayed on the surfaces of the model airplane and the wall. The design of Scene C for simultaneous 4D distance and velocity imaging is shown in Supplementary Fig. 4c. The upper part of the tunneled bridge of Scene A is set as the ROI, and the stationary models (two model trees and a traffic light) are located at intervals of 0.3 m from 6 m in the order mentioned. Similarly, a reflective spray was applied.”

Fig. 3.1. Fast Fourier transform (FFT) result of the interference signal by a single flutter-wavelength modulation. The interference signal of 42 dB signal-to-noise ratio (SNR) was obtained using a retroreflective target placed at 20 m.

The original beat frequency of the reflective tape at a distance of 20 m is shown in Fig. 3.1, which is shown in Fig. 2b. However, for clarity, we additionally noted the signal-to-noise ratio (SNR) in Fig. 2c with a description of the figure as follows:

Location: page 12, line 242

Location: page 13, line 253-255

“c Fast Fourier transform (FFT) result of the interference signal by a single EOPM modulation. The interference signal of 42 dB signal-to-noise ratio (SNR) was obtained using a retroreflective target placed at 20 m.”

As shown in Fig. 3.1, the SNR recorded approximately 42 dB for the reflective tape at a distance of 20 m. For the reflective spray-applied object at a 20 m distance, an SNR of 23 dB was acquired. Irrespective of material-dependent SNR variations, the notably high reflectivity remained consistent, allowing measurements up to 20 m at a given optical power. Since low optical power can be easily amplified using optical amplifiers, we expect that the imaging of normal objects can be achieved with sufficient power.

<Comment 4>

The achromatic lens used in this work can only ensure collimation within a short range. Can it still ensure that no obvious caustics occur at 20m?

<Response>

We appreciate your valuable comments. In our study, we employed a fiber collimator along with two achromatic lenses to manipulate the beam in free space. During each experiment, we adjusted the spacing between two lenses to achieve the focal of the beam at the desired distances. As the reviewer pointed out, in long-range three-dimensional (3D) distance measurements, a beam divergence due to the limitations of the working distance of the lenses is observed. However, it's worth noting that this divergence did not significantly compromise the integrity of the beam and had no significant impact on the measurement. We clarified the details in the revised manuscript as follows:

Location: page 24, line 503-504

“The spacing between the two lenses was adjusted to focus the beam at the desired distance for each experiment.”

<Comment 5>

What’s the modulation efficiency of the phase modulator that used in the FWSL? Why the proposed FWSL has good frequency-sweep modulation linearity? The unit of the amplitude of the triangular waveform in the supplementary material is also missing.

<Response>

We appreciate the reviewer for your thorough review of our manuscript and the insightful questions you’ve raised. First, the modulation efficiency⁴ of the utilized electro-optic phase modulator (EOPM) is $14000 V \cdot mm$ with respect to the center wavelength of FWSL.

Second, the high linearity during flutter-wavelength modulation is attributed to the electro-optical phase modulation. Within the FWSL cavity, changes in the refractive index occur in direct response to the modulation voltage applied to the EOPM, resulting in an optical path difference. Importantly, both the refractive index and optical path difference exhibit a linear relationship with the applied voltage. This optical path difference, in turn, affects the cavity of the FWSL, leading to precise and linear wavelength modulation. Furthermore, it’s worth noting that the EOPM supports modulation rates of up to 100 MHz, ensuring a robust and linear response even with a 100 kHz modulation.

Lastly, as the reviewer pointed out, we have modified the triangular waveform graph by adding the appropriate units. This modification has been reflected in Supplementary Fig. 12a, which is moved from Supplementary Fig. 1b. We hope these clarifications address your concerns, and we appreciate your meticulous attention to detail.

[4] Pathak, S. Photonics Integrated Circuits. in *Nanoelectronics* (ed. Kaushik, B. K.) 219–270 (Elsevier, 2019). doi:10.1016/B978-0-12-813353-8.00008-7.

Location: *Supplementary Information*

“

Supplementary Fig. 12. Linearity evaluation on flutter-wavelength modulation. (a) Raw interference signal; (b) Relative optical frequency behavior; Linear analysis including a regular residual plot of (c) up-chirp and (d) down-chirp."

<Comment 6>

Why does the frequency-sweep modulation bandwidth have the relationship with the cavity length of ECDL? This content that could have become advantages should be discussed and highlighted in the main text.

<Response>

We thank the reviewer for the thoughtful comments and sincere suggestions.

Fig. 6.1. Detailed structure of flutter-wavelength-swept laser (FWSL). AFG and WS denote arbitrary function generator and wavelength selector, respectively.

As shown in Fig. 6.1., our FWSL is an external cavity diode laser based on Littrow configuration⁵. In FWSL, the output wavelength is selected through a wavelength selector (WS), which consists of a mirror and a diffraction grating. The selected wavelength is modulated in a flutter shape (equal to the frequency sweep that the reviewer mentioned) and is achieved through cavity length modulation via EOPM. The mode-hop-free flutter-wavelength modulation bandwidth is limited by the free spectral range (*FSR*) of the cavity as⁵

$$|v_{EC} - v_g| \leq \frac{1}{2} FSR \quad (6.1)$$

where v_{EC} and v_g are the peak external cavity mode frequency within the diffraction grating reflection spectrum region and the peak reflection spectrum of the diffraction grating, respectively. *FSR* is determined as $FSR = \frac{c}{2L_{EC}}$, where c refers to the speed of light in vacuum and L_{EC} refers external cavity length of the FWSL. From the given relationship, it can be simply stated that the flutter-wavelength modulation bandwidth works inversely proportional to the cavity length change.

To demonstrate this relationship in more detail, we have conducted additional simulations and actual measurements. Referring to Fig. 6.1., a difference in the optical path, the cavity length, can be expected when the light incident on the diffraction grating is adjusted. In the Littrow configuration, the first-order feedback wavelength of the diffraction grating can be expressed as follows⁵,

$$\lambda_g = 2d \sin \theta \quad (6.2)$$

where d and θ are the grating period and the angle of light incident on the diffraction grating, respectively.

As introduced above, the L_{EC} changes when the wavelength is varied through WS and changing L_{EC} of the FWSL according to the output wavelength can be expressed as follows.

$$L_{EC} = L_{EC,1} + L_{EC,2} = L_{EC,1} + \frac{L_{AG}}{\cos \theta} \quad (6.3)$$

Thus, the FSR according to wavelength can be expressed as below.

$$FSR(\lambda) = \frac{c}{L_{EC,1} + \frac{L_{AG}}{\cos \left\{ \sin^{-1} \left(\frac{1}{2d\lambda} \right) \right\}}} \quad (6.4)$$

Fig. 5.2. Simulation results of differences in L_{EC} and mode-hop-free modulation range based on the output wavelength λ_g of FWSL.

It is expected that output wavelength sweep through WS will contribute to L_{EC} change, therefore, varying the mode-hop-free modulation range. The simulation results of the L_{EC} and mode-hop-free modulation range according to the λ_g are shown in Fig. 6.2 that are calculated with respect to the fabricated FWSL architecture ($L_{EC,1} \approx 150$ mm, $L_{AG} \approx 25$ mm, $d = 1050$ grooves/mm). As represented in Fig. 6.1., FWSL is designed such that L_{EC} extends as the λ_g increases. According to the present structure, the FWSL lengthens L_{EC} by approximately 11 mm, while the output wavelength sweeps from 1450 nm to 1650 nm. An

increase in L_{EC} affects inversely proportional to FSR so that the mode-hop free modulation range varies during the wavelength sweep.

Fig. 6.3. (a) Calculated and measured free spectral range (FSR) based on the wavelength; **(b)** Measured wavelength modulation bandwidth based on the wavelength with respect to a same electro-optic phase modulator (EOPM) operation.

Fig. 6.3a illustrates the calculated FSR and the measured FSR of the FWSL during the wavelength change. It is evident from the figure that the calculated and measured FSR changed equally depending on the wavelength, with similar reduction rates. The difference between the calculated and measured values is expected to occur during the component assembly and alignment processes of the FWSL. Fig. 6.3b shows the measured modulation bandwidth for each λ_g , which confirms that the modulation bandwidth tends to decrease as the λ_g increases.

In consideration of the reviewer's meticulous feedback, we have taken the opportunity to incorporate a more comprehensive explanation in the Supplementary Information, which is presented below.

[5] Haiping, G., Chenhao, W., Fei, W., Lijing, Z. & Chengwen, X. Study on the dynamic mode stability of grating-feedback external cavity diode lasers. *Laser Phys* **26**, 045002 (2016).

Location: *Supplementary Information*

“Wavelength modulation bandwidth

The mode-hop-free wavelength modulation range through a variable external cavity length is limited by the free spectral range (FSR) of the cavity, as shown below³:

$$|v_{EC} - v_g| \leq \frac{1}{2} FSR \quad (6)$$

where v_{EC} and v_g denote the peak external cavity mode frequencies within the diffraction grating reflection spectrum region and the peak reflection spectrum of the diffraction grating, respectively. FSR is determined as $FSR = \frac{c}{2L_{EC}}$, where c refers to the speed of light in vacuum.

Owing to the structural characteristics of the FWSL, the external cavity length changes when the wavelength is swept through the WS, as inferred from Supplementary Fig. 2. The external cavity length shown in Supplementary Fig. 2 can be expressed as follows:

$$L_{EC} = L_{EC,1} + L_{EC,2} = L_{EC,1} + \frac{L_{AG}}{\cos \theta} \quad (7)$$

Accordingly, FSR can be expressed as

$$FSR(\lambda) = \frac{c}{L_{EC,1} + \frac{L_{AG}}{\cos \left\{ \sin^{-1} \left(\frac{1}{2d\lambda} \right) \right\}}} \quad (8)$$

Supplementary Fig. 6. Simulation results of differences in L_{EC} and mode-hop-free modulation range based on the output wavelength λ_g of FWSL.

L_{EC} and mode-hop free modulation range based on the output wavelength of the FWSL simulated reflecting the specifications of the fabricated FWSL— $L_{EC,1} \approx 150 \text{ mm}$, $L_{AG} \approx 25 \text{ mm}$, $d = 1050 \text{ grooves/mm}$ —are shown in Supplementary Fig. 6. As shown in Supplementary Fig. 2, the FWSL was designed such that L_{EC} increases as the reflected wavelength of the WS increases. According to the present architecture, the FWSL lengthens L_{EC} by approximately 11 mm, while the output wavelength sweeps from 1450 to 1650 nm. An increase in L_{EC} decreases FSR , and the mode-hop-free modulation range also decreases, according to Eq. 6.

Supplementary Fig. 7. Simulation results of the output wavelength change based on the same cavity length change at different λ_g .

Supplementary Table 3. Calculation results of FSR , $Slope$, and relative ratios of $Slope$ according to the wavelength.

Wavelength [nm]	1480	1500	1520	1540	1560	1580	1600
FSR [GHz]	0.7991	0.7955	0.7916	0.7875	0.783	0.7782	0.7729
Slope [GHz/nm]	0.00108	0.001061	0.001042	0.001023	0.001004	0.000985	0.000966
Norm. slope [%]	100	98.235	96.481	94.732	92.984	91.233	89.486

Supplementary Fig. 7 illustrates the simulation results based on Eq. 4, which shows the change in the output wavelength when the cavity length is varied. The output wavelength change ($Slope$) according to the optical path difference can be expressed as

$$Slope(\lambda) = \frac{2 \cdot FSR(\lambda)}{\lambda} \quad (9)$$

Supplementary Table 3 contains the results of calculated FSR , $Slope$, and the relative ratio of $Slope$ for each wavelength. For the calculation, the FSR used the values derived from Supplementary Fig. 6. As a result of the calculations, the output wavelength change ($Slope$)

according to the varying cavity length tended to decrease by around 89% as the center wavelength increased.

Supplementary Fig. 8. (a) Calculated and measured *FSR* based on the wavelength; **(b)** Measured wavelength modulation bandwidth based on the wavelength with respect to a same electro-optic phaser modulator (EOPM) operation.

Supplementary Fig. 8a shows the calculated *FSR* and measured *FSR* of the FWSL. It is evident from the figure that the calculated and measured *FSR* changed equally depending on the wavelength, with similar reduction rates. The difference between the calculated and measured values is expected to occur during the component assembly and alignment processes of the FWSL.

Supplementary Table 4. Measured modulation bandwidth and relative ratios based on the wavelength.

Wavelength [nm]	1480	1500	1520	1540	1560	1580	1600
Modulation bandwidth [GHz]	0.74273	0.73303	0.71968	0.69843	0.68858	0.67177	0.65923
Norm. modulation bandwidth [%]	100	98.694	96.896	94.035	92.709	90.446	88.757

Supplementary Fig. 8b shows the measured modulation bandwidth for each wavelength. As confirmed by Eq. 9 and Supplementary Table 3, the actual modulation bandwidth tends to decrease as the selected wavelength increases. Supplementary Table 4 depicts the measured modulation bandwidth for each wavelength and the standard relative ratio. In particular, the norm. modulation bandwidth in Supplementary Table 4 shows a tendency similar to that of the norm. slope and the theoretical analysis values in Supplementary Table 3.”

[3] Haiping, G., Chenhao, W., Fei, W., Lijing, Z. & Chengwen, X. Study on the dynamic mode stability of grating-feedback external cavity diode lasers. *Laser Phys* **26**, 045002 (2016).

<Comment 7>

0.21m ranging resolution may be reasonable to the long-range detection, but how to solve the problem of poor range resolution of close range targets?

<Response>

Thank you for the detailed review and for raising an important question. As the reviewer has pointed out, a proper ranging resolution must be secured for practical LiDAR applications. For example, in automotive applications, a ranging resolution of approximately 5 cm is recommended⁶. Only considering the axial resolution, 21 cm, determined by our FWSL, it may not be sufficient. However, LiDAR is a technology for topography over tens and hundreds of m-scale, of which the practical distance resolution for the measurements should be considered as the precision of the overall system. From the point of view, our system supports distance resolution under 4 cm as experimentally demonstrated through axial coherent ranging up to 25 m. The practical velocity resolution of around 6 cm/s is shown through velocity measurement on the rotating disk. Moreover, it is again experimentally demonstrated in Supplementary Video 1, which shows clearly distinct four-dimensional (4D) imaging for objects at relatively short distances without any interruption.

However, we admit that improvement of the axial resolution by FWSL is required, which may affect the accuracy of the system. As mentioned in the discussion section previously, the axial resolution can surely be improved by increasing the flutter-modulation bandwidth through careful optimization of the architecture of FWSL with the improvement of equipment.

We appreciate the sincere concern of the reviewer and have clearly defined the distance and velocity resolutions for practical measurements as follows.

[6] Holzhüter, H., Bödewadt, J., Bayesteh, S., Aschinger, A. & Blume, H. Technical concepts of automotive LiDAR sensors: a review. *Optical Engineering* **62**, (2023).

Location: page 22, line 439-446

“However, the axial resolution does not necessarily define the distance measurement resolution of our LiDAR system. Regarding LiDAR is a technology for topography over m-scale, the practical distance resolution and velocity resolution for the measurements should be considered as the precision of the overall system. As demonstrated above, our system supports distance resolution under 0.04 m and velocity resolution around 0.06 m/s. *Supplementary Video 1*, which shows a distinct 4D imaging of the objects without any interruption, experimentally proves that the actual measurements are more precise than the axial resolution.”

[Reviewer 2]

D. Jeong and colleagues present a real time coherent distance and velocity measurement system based on the modulation of a semiconductor external cavity diode laser using a conventional grating pivot mechanism for coarse tuning and an intracavity phase modulator for fast fine tuning. Hence the beam scanning and the ranging modulation in spectral-spatial LiDAR using the flutter-wavelength-swept laser can be decoupled to improve performance. This paper solves one of the issues of earlier works by Okano et al. and achieves a 90 kHz pixel measurement rate, 10 Hz frame rate, 750 MHz flutter chirp bandwidth, corresponding to a 20 cm resolution, with a power of 5 mW power on the transmission grating aperture. The paper is in general well written and the figures are with some minor exceptions well crafted. Overall, I support publication in Nature Communications but would ask the authors to answer the following questions and comments:

<Comment 1>

Generally the authors should give more information on the FWSL, which is the core innovation of the paper. What are the free-spectral range of the laser, which should limit the optical linewidth? What limitations exist on the EO tuning range? I think there should be a trade-off between the fast-tuning range and the optical linewidth. Careful simulations of the relationship between laser linewidth showed us that a linewidth of less than 50 kHz is desirable for long range LiDAR beyond 150m. Can this be achieved while also achieving a cm-scale distance resolution with the current FWSL architecture? What is the architecture of the laser (pivot mechanism or simple Littrow grating)? Is the coarse wavelength sweep mode hop free?

<Response>

We appreciate the reviewer for your meticulous review and the insightful questions you've raised.

Fig. 1.1. Detailed schematic diagram of the flutter-wavelength-swept laser (FWSL). AFG and WS denote arbitrary function generator and wavelength selector, respectively.

The flutter-wavelength-swept laser (FWSL) proposed in this paper has an external cavity diode laser (ECDL) structure. The structure is based on Littrow configuration with a tunable reflective filter a wavelength selector (WS), which consists of a mirror and a reflective holographic diffraction grating¹. The detailed architecture of the FWSL is provided in Fig. 1.1.

The laser's free spectral range (*FSR*) is identified by an external cavity length (L_{EC}) as below.

$$FSR = \frac{c}{2L_{EC}} \quad (1.1)$$

where c is the speed of light in vacuum and L_{EC} is the external cavity length of the FWSL. The flutter-wavelength modulation is achieved through cavity length modulation via an electro-optic phase modulator (EOPM). The mode-hop-free modulation range of EOPM is limited by the *FSR* below¹.

$$|v_{EC} - v_g| \leq \frac{1}{2} FSR \quad (1.2)$$

Here, v_{EC} and v_g are the peak external cavity mode frequency within the diffraction grating reflection spectrum region and the peak reflection optical frequency of the diffraction grating, respectively.

As introduced above, the L_{EC} changes when the wavelength is varied through WS, which can be inferred from Fig. 1.1. L_{EC} of the FWSL can be expressed as follows.

$$L_{EC} = L_{EC,1} + L_{EC,2} = L_{EC,1} + \frac{L_{AG}}{\cos \theta} \quad (1.3)$$

Finally, the *FSR* according to wavelength can be expressed as below.

$$FSR(\lambda) = \frac{c}{L_{EC,1} + \frac{L_{AG}}{\cos \left\{ \sin^{-1} \left(\frac{1}{2d\lambda} \right) \right\}}} \quad (1.4)$$

Fig. 1.2. Simulation results of differences in L_{EC} and mode-hop-free tuning range according to the output wavelength λ_g of the FWSL.

The simulation results of the L_{EC} and mode-hop-free tuning range according to the λ_g are shown in Fig. 1.2.. The parameters for the simulations are set as $L_{EC,1} \simeq 150$ mm, $L_{AG} \simeq 25$ mm, and $d = 1050$ grooves/mm. As represented in Fig. 1.1, FWSL was designed such that L_{EC} increases as the reflected wavelength of the WS increases. According to the present architecture, the FWSL lengthens L_{EC} by approximately 11 mm, while the output wavelength sweeps from 1450 to 1650 nm. An increase in L_{EC} decreases FSR , and the mode-hop-free modulation range also decreases, according to Eq. 1.2.

Fig. 1.3. Simulation results of output wavelength change according to the same cavity length change at different λ_g .

Supplementary Fig. 7 illustrates the simulation results based on Eq. 4, which shows the change in the output wavelength when the cavity length is varied. The output wavelength change (*Slope*) according to the optical path difference can be expressed as

$$Slope(\lambda) = \frac{2 \cdot FSR(\lambda)}{\lambda} \quad (1.5)$$

Table 1.1 contains the results of calculated *FSR*, *Slope*, and the relative ratio of *Slope* for each wavelength. For the calculation, the *FSR* used the values derived from Supplementary Fig. 1.2. As a result of the calculations, the output wavelength change (*Slope*) according to the varying cavity length tended to decrease by around 89% as the center wavelength increased.

Table 1.1. Calculation results of *FSR*, *Slope*, and relative ratios of *Slope* according to wavelength.

Wavelength [nm]	1480	1500	1520	1540	1560	1580	1600
FSR [GHz]	0.7991	0.7955	0.7916	0.7875	0.783	0.7782	0.7729
Slope [GHz/nm]	0.00108	0.001061	0.001042	0.001023	0.001004	0.000985	0.000966
Norm. slope [%]	100	98.235	96.481	94.732	92.984	91.233	89.486

Fig. 1.4. (a) Calculated and measured free spectral range (*FSR*) according to the wavelength; **(b)** Measured wavelength modulation bandwidth by electro-optic phase modulator (EOPM) according to the wavelength.

Fig. 1.4a shows the calculated *FSR* and measured *FSR* of the FWSL. It is evident from the figure that the calculated and measured *FSR* changed equally depending on the wavelength, with similar reduction rates. The difference between the calculated and measured values is expected to occur during the component assembly and alignment processes of the FWSL.

Table 1.2. Measured modulation bandwidth and relative ratios according to wavelength.

Wavelength [nm]	1480	1500	1520	1540	1560	1580	1600
Modulation bandwidth [GHz]	0.74273	0.73303	0.71968	0.69843	0.68858	0.67177	0.65923
Norm. modulation bandwidth [%]	100	98.694	96.896	94.035	92.709	90.446	88.757

Fig. 1.4b shows the measured modulation bandwidth for each wavelength. As confirmed by Eq. 1.5 and Table 1.1., the actual modulation bandwidth tends to decrease as the selected wavelength increases. Table 1.2. depicts the measured modulation bandwidth for each wavelength and the standard relative ratio. In particular, the norm. modulation bandwidth in Table 1.2. shows a tendency similar to that of the norm. slope and the theoretical analysis values in Table 1.1..

Fig. 1.5. Measured linewidth with the Lorentz fit. HWHM denotes half-width at half-maximum.

As such, the modulation bandwidth of the fabricated FWSL decreases with increasing wavelength, eventually reaching approximately 0.66 GHz (at 1600 nm). The corresponding axial resolution is about 23 cm, which is not very high. However, it is due to the current architecture of FWSL, which indicates that there remains room for further improvement. Based on the complex relationships above, a wider wavelength modulation bandwidth for cm-level distance resolution can surely be achieved through careful optimization of the cavity architecture. Together, the narrow linewidth, which benefits from the ECDL structure itself, is expected to be achieved independently, along with the wider wavelength modulation bandwidth. After a more accurate optical alignment of FWSL, the remeasured optical linewidth is 81 kHz, as shown in Fig. 1.5. It shows an improvement over the previous linewidth of 124 kHz, which implies there is still room for improvement for narrower linewidths.

Fig. 1.6 (a)-(d) Principle of ambiguous distance formation as the optical path difference increases; **(e)-(h)** Beat frequency, f_b , change aspects based on the distance change ($d_2 > d_1$) at each condition.

Considering the nature of frequency-modulated continuous-wave (FMCW) LiDAR, which utilizes a frequency-modulated signal with periodicity, the distance that can be measured without ambiguity is not only affected by coherence length but also determined by the modulation speed. When the wavelength is modulated using a 50% symmetry triangle waveform, the ambiguous distance is determined as

$$D_{amb} = \frac{c}{2f_{sweep}} \quad (2.1.6.)$$

where c and f_{sweep} refer to the speed of light in vacuum and the flutter-wavelength modulation rate, respectively.

Fig. 1.6 shows the beat frequency (f_b) formation and change aspects for each distance condition of the measurement target. First, Fig. 1.6a, e shows the formation of f_b in the area where distance can be normally measured ($0 \leq d < D_{amb}$). Both the reference signal (S_{ref}) and the reflected signal (S_{sig}) that form f_b in the corresponding area are of the same order, and therefore, as the distance increases ($d_1 \rightarrow d_2$), f_b also increases proportionally. Fig. 1.6b and f shows the formation of f_b when the position of the target is equal to the ambiguous distance ($d = D_{amb}$). As shown in the figure, f_b is not measured because there is no simultaneously overlapping area between chirping signals in the same direction, f_b is not measured, as seen from the figure. Fig. 1.6c, g illustrates the case when the condition $0 \leq d < 2D_{amb}$ holds. In this case, $S_{sig,N}$ forms an interference signal with $S_{ref,N+1}$, which is the next-order reference

signal and f_b decreases as distance increases. This phenomenon can be overcome using complex conjugate resolutions². Finally, Fig. 1.6d and h show that f_b increases proportionally as the distance increases; however, when the order (N) between the reference and the reflected signals differs by 1, it can be observed that Fig. 1.6d and h have the same shape as Fig. 1.6a and e because of ambiguity. Therefore, an optimal modulation speed should be set to prevent such occurrences of ambiguity, regardless of the optical linewidth.

Lastly, for robust and reliable beat signal generation, mode-hop-free operation is important. Previously, the mode-hop-free condition in the external cavity during the flutter-wavelength modulation was confirmed in Eq. 1.2. Unfortunately, for the wavelength sweep through WS, mode hopping inevitably occurs because the mode order of the external cavity changes. However, as already proven from our measurement results, such mode hopping during the wavelength sweep does not appear to be critical to the overall four-dimensional (4D) image acquisition.

Once more, we would like to express our sincere gratitude for the reviewer's comments. In the interest of enhancing the value of our work for our readers, we are pleased to incorporate these analyses into the supplementary information, as detailed below.

[1] Haiping, G., Chenhao, W., Fei, W., Lijing, Z. & Chengwen, X. Study on the dynamic mode stability of grating-feedback external cavity diode lasers. *Laser Phys* **26**, 045002 (2016).

[2] Sarunic, M. V., Choma, M. A., Yang, C. & Izatt, J. A. Instantaneous complex conjugate resolved spectral domain and swept-source OCT using 3x3 fiber couplers. *Opt Express* **13**, 957 (2005).

Location: *Supplementary Information*

“FWSL structure and principles

The detailed structure of the FWSL is depicted in Supplementary Fig. 2. The FWSL adopts a Littrow configuration with a tunable reflective filter structure based on a wavelength selector (WS), which consists of a mirror and a reflective holographic diffraction grating (1050 grooves/mm). The broadband light generated from the gain is collimated by the collimator and passed through the electro-optic phase modulator (EOPM), which is for flutter-wavelength modulation. Then, an output wavelength is selected via the WS by adjusting the angle of light incident on the diffraction grating. The reflected light of the selected wavelength is lased within the cavity from the gain to the WS and is emitted through a fiber-coupled output. During the overall operation, the output wavelength sweep via WS and the flutter-wavelength modulation via EOPM are synchronized using an electrical control signal from an arbitrary function generator.

Supplementary Fig. 7. Detailed structure of FWSL. AFG and WS denote arbitrary function generator and wavelength selector, respectively.

The output wavelength of the FWSL is determined by a combination of the cavity spectrum of the gain chip, the longitudinal mode spectrum of the external cavity, and the reflection spectrum of the WS. The internal cavity gain of the gain chip can be described as follows^{3,4}:

$$G_{IC} = \frac{G}{(1 + G\sqrt{R_1R_2})^2 + 4G\sqrt{R_1R_2} \sin^2 \theta} \quad (1)$$

where G denotes the single pass gain, R_1 , R_2 indicate the reflectance of both facets of the gain chip, and θ refers to the phase difference that can be expressed as $\theta = 4\pi L_{IC}/\lambda$, where L_{IC} and λ denote the internal cavity length and wavelength⁴, respectively. The external cavity gain is expressed as⁴

$$G_{EC} = \frac{T_2}{[1 - G\sqrt{R_1T_2R_3}\sqrt{R_4}]^2 + 4G\sqrt{R_1T_2R_3}\sqrt{R_4} \sin^2 \delta} \quad (2)$$

where R_3 and R_4 indicate the reflectance of the angle scanner and diffraction grating, respectively, and δ denotes the phase difference, which has a relation of $\theta = 4\pi L_{EC}/\lambda$ with external cavity length, L_{EC} . The reflectance of a diffraction grating can be expressed as⁵

$$R_4 = R_g \left(\frac{\sin(N\phi/2)}{N \sin(\phi/2)} \right)^2 \quad (3)$$

where R_g denotes the peak reflectance of the grating, N corresponds to the illuminated groove number of the diffraction grating, and ϕ symbolizes the phase difference, which can be calculated by $\phi = 2m\pi\lambda_0/\lambda$, where λ_0 indicates the feedback wavelength of the grating. The

gain spectrum of the FWSL is determined by the product of the above factors expressed as follows⁴:

$$G_{\text{net}} = R_4 G_{IC} G_{EC} \quad (4)$$

For simplicity, the losses of the collimator and the EOPM are ignored.

Supplementary Fig. 3 shows the structural configuration of the cavity mode of the FWSL. The cavity configuration parameters for the simulation are listed in Supplementary Table 2. Based on these simulations, three identical lasers were fabricated, which operated well as intended, demonstrating the designed features of the FWSL.

Supplementary Fig. 8. Simulation results of the cavity mode structure of the FWSL.

Supplementary Table 2. Parameters and values for the cavity spectrum simulation.

Parameter	L_{IC} (mm)	L_{EC} (mm)	G	R_1	R_2 (T_2)	R_3	R_g	λ_g (nm)	N
Value	4.65	50.1	1	0.1	0.005 (0.995)	1	1	1550	10000

Principle of wavelength change

Previously, it was confirmed that the output wavelength of the FWSL is determined by a complex spectrum of the internal and external cavities, and each cavity gain is derived from the cavity length. The external cavity, which exhibits a relatively high gain, primarily influences the wavelength selection. In other words, the output wavelength can be changed by adjusting the length of the external cavity in the FWSL⁵. In addition, the output wavelength can be changed by varying the angle of the light incident on the diffraction grating³.

Supplementary Fig. 9. (a) Simulation results of G_{EC} according to L_{EC} change; (b) Resulting peak wavelength change.

The FWSL utilizes both techniques for wavelength changes. First, flutter-wavelength modulation was achieved by controlling the external cavity length through EOPM at each fixed angle of the incident light on the diffraction grating. Supplementary Fig. 4 illustrates the simulation results of the different gain spectrum when the external cavity length was changed. The remaining parameters, except L_{EC} used the same values as those in Supplementary Table 2.

Next, a wavelength sweep using the WS was implemented by changing the angle of light incident on the diffraction grating. In the Littrow configuration, the first-order feedback wavelength of the diffraction grating can be expressed as³

$$\lambda_g = 2d \sin \theta \quad (5)$$

where d and θ denote the grating period and the angle of light incident on the diffraction grating, respectively. Based on the gain spectrum of the external cavity (Eq. 2) along with the effect of the diffraction grating, only external cavity modes of a certain order are selected and survive, depending on the angle of light incident on the diffraction grating. Supplementary Fig. 5 presents the simulation results of the different gain spectrum when λ_g changes. For the simulation, the remaining parameters, except for λ_g , had the same values as those listed in Supplementary Table 2.

Supplementary Fig. 10. (a) Simulation results of G_{EC} based on the change in λ_g ; (b) Resulting peak wavelength change.

Wavelength modulation bandwidth

The mode-hop-free wavelength modulation range through a variable external cavity length is limited by the free spectral range (FSR) of the cavity, as shown below³:

$$|v_{EC} - v_g| \leq \frac{1}{2} FSR \quad (6)$$

where v_{EC} and v_g denote the peak external cavity mode frequencies within the diffraction grating reflection spectrum region and the peak reflection spectrum of the diffraction grating, respectively. FSR is determined as $FSR = \frac{c}{2L_{EC}}$, where c refers to the speed of light in vacuum.

Owing to the structural characteristics of the FWSL, the external cavity length changes when the wavelength is swept through the WS, as inferred from Supplementary Fig. 2. The external cavity length shown in Supplementary Fig. 2 can be expressed as follows:

$$L_{EC} = L_{EC,1} + L_{EC,2} = L_{EC,1} + \frac{L_{AG}}{\cos \theta} \quad (7)$$

Accordingly, FSR can be expressed as

$$FSR(\lambda) = \frac{c}{L_{EC,1} + \frac{L_{AG}}{\cos \left\{ \sin^{-1} \left(\frac{1}{2d\lambda} \right) \right\}}} \quad (8)$$

Supplementary Fig. 11. Simulation results of differences in L_{EC} and mode-hop-free modulation range based on the output wavelength λ_g of FWSL.

L_{EC} and mode-hop free modulation range based on the output wavelength of the FWSL simulated reflecting the specifications of the fabricated FWSL— $L_{EC,1} \approx 150 \text{ mm}$, $L_{AG} \approx 25 \text{ mm}$, $d = 1050 \text{ grooves/mm}$ —are shown in Supplementary Fig. 6. As shown in Supplementary Fig. 2, the FWSL was designed such that L_{EC} increases as the reflected wavelength of the WS increases. According to the present architecture, the FWSL lengthens L_{EC} by approximately 11 mm, while the output wavelength sweeps from 1450 to 1650 nm. An increase in L_{EC} decreases FSR , and the mode-hop-free modulation range also decreases, according to Eq. 6.

Supplementary Fig. 7. Simulation results of the output wavelength change based on the same cavity length change at different λ_g .

Supplementary Table 3. Calculation results of FSR , $Slope$, and relative ratios of $Slope$ according to the wavelength.

Wavelength [nm]	1480	1500	1520	1540	1560	1580	1600
FSR [GHz]	0.7991	0.7955	0.7916	0.7875	0.783	0.7782	0.7729
Slope [GHz/nm]	0.00108	0.001061	0.001042	0.001023	0.001004	0.000985	0.000966
Norm. slope [%]	100	98.235	96.481	94.732	92.984	91.233	89.486

Supplementary Fig. 7 illustrates the simulation results based on Eq. 4, which shows the change in the output wavelength when the cavity length is varied. The output wavelength change ($Slope$) according to the optical path difference can be expressed as

$$Slope(\lambda) = \frac{2 \cdot FSR(\lambda)}{\lambda} \quad (9)$$

Supplementary Table 3 contains the results of calculated FSR , $Slope$, and the relative ratio of $Slope$ for each wavelength. For the calculation, the FSR used the values derived from Supplementary Fig. 6. As a result of the calculations, the output wavelength change ($Slope$)

according to the varying cavity length tended to decrease by around 89% as the center wavelength increased.

Supplementary Fig. 8. (a) Calculated and measured *FSR* based on the wavelength; **(b)** Measured wavelength modulation bandwidth based on the wavelength with respect to a same electro-optic phaser modulator (EOPM) operation.

Supplementary Fig. 8a shows the calculated *FSR* and measured *FSR* of the FWSL. It is evident from the figure that the calculated and measured *FSR* changed equally depending on the wavelength, with similar reduction rates. The difference between the calculated and measured values is expected to occur during the component assembly and alignment processes of the FWSL.

Supplementary Table 4. Measured modulation bandwidth and relative ratios based on the wavelength.

Wavelength [nm]	1480	1500	1520	1540	1560	1580	1600
Modulation bandwidth [GHz]	0.74273	0.73303	0.71968	0.69843	0.68858	0.67177	0.65923
Norm. modulation bandwidth [%]	100	98.694	96.896	94.035	92.709	90.446	88.757

Supplementary Fig. 8b shows the measured modulation bandwidth for each wavelength. As confirmed by Eq. 9 and Supplementary Table 3, the actual modulation bandwidth tends to decrease as the selected wavelength increases. Supplementary Table 4 depicts the measured modulation bandwidth for each wavelength and the standard relative ratio. In particular, the norm. modulation bandwidth in Supplementary Table 4 shows a tendency similar to that of the norm. slope and the theoretical analysis values in Supplementary Table 3.

Mode hopping

Previously, the mode-hop-free condition in the external cavity during wavelength modulation through EOPM was confirmed using Eq. 6. However, mode hopping may occur if the initial

state has a sufficiently large difference between the external cavity mode and central reflection wavelength of the diffraction grating. Furthermore, unexpected mode hopping phenomenon is believed to be caused by the asymmetric nonlinear gain⁶⁻⁹. However, the mode hopping phenomenon could hardly be observed in the actual experimental process during 2 years of expected lifetime of fabricated FWSL.

During the wavelength sweep through a WS, mode hopping inevitably occurs because the mode order of the external cavity changes. Mode hopping can be suppressed using a pivot-based external cavity structure¹⁰ or quasi-phase continuous tuning¹¹ that matches the reflected wavelength of the diffraction grating and the external cavity mode. However, in the proposed FWSL, the WS and EOPM operate sequentially; therefore, the fixed-order cavity mode is assured during flutter-wavelength modulation through the EOPM. In addition, the measured modulation bandwidth (Supplementary Fig. 8b, Supplementary Table 4) is within the maximum mode-hop-free modulation range (Supplementary Fig. 8a, Supplementary Table 3), and there exists hysteresis⁶⁻⁹. It is expected that mode hopping during the EOPM driving process will rarely occur, which can be fatal in axial coherent ranging.”

[3] Haiping, G., Chenhao, W., Fei, W., Lijing, Z. & Chengwen, X. Study on the dynamic mode stability of grating-feedback external cavity diode lasers. *Laser Phys* **26**, 045002 (2016).

[4] Gong, H., Liu, Z., Zhou, Y. & Zhang, W. Extending the mode-hop-free tuning range of an external-cavity diode laser by synchronous tuning with mode matching. *Appl Opt* **53**, 7878 (2014).

[5] Levin, L. Mode-hop-free electro-optically tuned diode laser. *Opt Lett* **27**, 237 (2002).

[6] Bogatov, A., Eliseev, P. & Sverdlov, B. Anomalous interaction of spectral modes in a semiconductor laser. *IEEE J Quantum Electron* **11**, 510–515 (1975).

[7] Ogasawara, N. & Ito, R. Longitudinal Mode Competition and Asymmetric Gain Saturation in Semiconductor Injection Lasers. II. Theory. *Jpn J Appl Phys* **27**, 615 (1988).

[8] Yamada, M. Theoretical analysis of nonlinear optical phenomena taking into account the beating vibration of the electron density in semiconductor lasers. *J Appl Phys* **66**, 81–89 (1989).

[9] F. N. Timofeev M. S. Shatalov S. A. Gurevich P. Bayvel R. Wyatt I. Lealman R. Kashyap, G. S. S. Experimental and Theoretical Study of High Temperature-Stability and Low-Chirp 1.55 μm Semiconductor Laser with an External Fiber Grating. *Fiber and Integrated Optics* **19**, 327–353 (2000).

[10] Trutna, W. R. & Stokes, L. F. Continuously tuned external cavity semiconductor laser. *Journal of Lightwave Technology* **11**, 1279–1286 (1993).

[11] Chong, C., Suzuki, T., Morosawa, A. & Sakai, T. Spectral narrowing effect by quasi-phase continuous tuning in high-speed wavelength-swept light source. *Opt Express* **16**, 21105 (2008).

<Comment 2>

Why is the laser linewidth measurement with the self-heterodyne interferometer (see Supplementary Figure 1a) so noisy? What was the fiber delay length in the interferometer chosen for the measurement?

<Response>

Thank you for your kind comments and requesting pertinent information. The previous linewidth data were obtained by setting a relatively low noise-filtering bandwidth for the spectrum analyzer, which resulted in noise. The linewidth was remeasured by applying the appropriate settings of the spectrum analyzer, as shown below:

Location: *Supplementary Information*

Supplementary Fig. 10. Measured linewidth with the Lorentz fit. HWHM denotes half-width at half-maximum.

Fig. 2.1. Experimental schematic of self-heterodyne interferometer for linewidth

measurement.

To measure the optical linewidth, we used a self-heterodyne interferometer with an SMF-28 fiber delay line of 10 km length. A schematic of the interferometer^{3,4} is illustrated in Fig. 2.1. The length of the fiber delay line was sufficiently longer than the coherence length. The remeasured linewidth of the FWSL was 81 kHz, which corresponds to a coherence length of approximately 1.2 km^{3,4}. We modified the Supplementary Information based on the updated results as follows:

[3] Okoshi, T., Kikuchi, K. & Nakayama, A. Novel method for high resolution measurement of laser output spectrum. *Electron Lett* **16**, 630 (1980).

[4] Mercer, L. B. 1/f frequency noise effects on self-heterodyne linewidth measurements. *Journal of Lightwave Technology* **9**, 485–493 (1991).

Location: *Supplementary Information*

“The optical linewidth of the FWSL was measured to estimate the coherence length. A self-heterodyne interferometer^{12,13} with a 10 km fiber delay line was used for the measurement. An electro-optic modulator was used as a frequency shifter driven by a 25 MHz sine function. The center wavelength of the FWSL was set at 1535 nm, and the output spectrum was measured using a spectrum analyzer as shown in Supplementary Fig. 9. The half-width at half-maximum (HWHM) of the Lorentz fitting of the measured spectrum was 81 kHz. The corresponding coherence length is approximately 1.2 km^{12,13}.”

[12] Okoshi, T., Kikuchi, K. & Nakayama, A. Novel method for high resolution measurement of laser output spectrum. *Electron Lett* **16**, 630 (1980).

[13] Mercer, L. B. 1/f frequency noise effects on self-heterodyne linewidth measurements. *Journal of Lightwave Technology* **9**, 485–493 (1991).

<Comment 3>

The V_{pi} modulation voltage requirement of the EOPM does increase with wavelength. How does the flutter modulation bandwidth with the EOPM change with respect to laser wavelength given the relative bandwidth of 10% of the laser sweep? It seems the authors have done all the measurements with a constant flutter modulation voltage. How large would the ranging error be?

<Response>

We thank the reviewer for the thoughtful comments and questions. Throughout the experiments, we maintained a constant flutter-modulation signal with an amplitude of 200 V_{pp} to the EOPM. As the reviewer pointed out, the required value of the V_{pi} modulation voltage of the EOPM increases with the wavelength. The V_{pi} requirements for the used EOPM are as Fig. 3.1.

Fig. 3.1. V_{pi} requirement along the wavelength from 1530 nm to 1540 nm.

The flutter-wavelength modulation bandwidth is affected by the cavity length, which is changed by EOPM. The cavity length change dL depending on the voltage V_{in} applied to EOPM and V_{pi} can be expressed based on the following relation⁵:

$$dL = \frac{\lambda V_{in}}{2V_{pi}} \quad (2.3.1.)$$

Within an approximate 10% sweep bandwidth spanning from 1530 nm to 1540 nm, relative to the center wavelength of 1535 nm, the observed error rates stand at 0.04% and 0.24%, respectively. It is important to note that these corresponding error rates are not expected to generate significant errors in overall coherent ranging.

[5] Boggs, B., Greiner, C., Wang, T., Lin, H. & Mossberg, T. W. Simple high-coherence rapidly tunable external-cavity diode laser. *Opt Lett* **23**, 1906 (1998).

<Comment 4>

What is the output power of the laser? Do the authors require an amplifier to achieve imaging without the reflective tape? What is the SNR of the beat notes on the 20 m non-reflective wall?

<Response>

We sincerely appreciate your kind review and the important questions you've raised. The output power of the FWSL was approximately 20 mW. To provide clarity regarding the FWSL power, we added a description in the manuscript as follows:

Location: page 24, line 497

“The optical power of the FWSL at 1535 nm was measured to be 20 mW.”

In our proposed system, the optical power decreases to 5 mW, passing through the interferometer and optical system. As it exhibits relatively low optical power, it was not able to obtain a beat frequency of a 20 m non-reflective wall. Therefore, to overcome this limitation, a reflective spray or tape was applied to the targets to enhance the reflectivity to obtain clear images during the experiments.

It is worth noting that the original beat frequency of the reflective tape at a distance of 20 m is provided in Fig. 2c. It was obtained with the same optical power of 5 mW under the same methodology as in the previous experiments. We indicated an observed signal-to-noise (SNR) of 42 dB in the figure, with the mention of its description as below.

Location: page 12, line 242

Location: page 13, line 253-255

“c Fast Fourier transform (FFT) result of the interference signal by a single EOPM modulation. The interference signal of 42 dB signal-to-noise ratio (SNR) was obtained using a retroreflective target placed at 20 m.”

The SNR of the reflective spray-applied object at a distance of 20 m was obtained as 23 dB. Irrespective of material-dependent SNR variations, the remarkably high reflectivity remained consistent, allowing the measurements up to 20 m at a given optical power. Since low optical power can be easily amplified using optical amplifiers, we expect that the imaging of normal objects can be achieved with sufficient power.

We admit that the insufficient explanation of the target configuration may have confused readers. Therefore, we detailed the target configuration in the revised manuscript.

Location: page 26, line 523-534

“Target configuration for 4D coherent ranging

The design of Scene A for room-scale 3D distance imaging is illustrated in Supplementary Fig. 4a. The first bus, police car, tunneled bridge, and last white screen are located at 4.8 m, 5.4 m, 6 m, and 7.3 m from the TG, respectively. A reflective spray (ALBEDO 100, Sweden) was applied to the surface to enhance the reflectivity of the targets. The design of Scene B for long-range 3D distance imaging is depicted in Supplementary Fig. 4b. The first box on the far right, second, third, fourth boxes, and last wall are located at 13.5 m, 16 m, 18.5 m, 21 m, and 22.5 m from the TG, respectively. Reflective tape (3M, Saint Paul, MN, USA) was applied to the boxes to enhance reflectivity, and a reflective spray was sprayed on the surfaces of the model airplane and the wall. The design of Scene C for simultaneous 4D distance and velocity imaging is shown in Supplementary Fig. 4c. The upper part of the tunneled bridge of Scene A is set as the ROI, and the stationary models (two model trees and a traffic light) are located at intervals of 0.3 m from 6 m in the order mentioned. Similarly, a reflective spray was applied.”

<Comment 5>

Is the metric for linearity the RMS deviation or the maximum error? Please also plot the deviation from linear fit in Figure 1b or in the supplement. It is impossible for the reader to verify the strong claim of 0.99997 linearity based on the presented data. Plotting the chirps as the relative wavenumber is quite unusual in the field of coherent LiDAR, where usually a frequency axis is chosen.

<Response>

Thank you for your careful review of our manuscript with valuable suggestions. All values mentioned as ‘linearity’ were R-squared values, and we apologize for inaccurate expressions for the linearity standard. All parts written as ‘linearity’ have been replaced with the expression ‘R-squared value,’ as shown below.

Location: page 10, line 199-200

“The R-squared values in each linear variable period for up-chirp and down-chirp are 0.99997 and 0.99987, respectively.”

Location: page 13, line 252-253

“The R-squared values of the up-chirp and down-chirp modulation are 0.99997 and 0.99987, respectively.”

In response to the reviewer’s valuable suggestions, we have included the fitted curve and the corresponding regular residual graph of the linear regression analysis in Supplementary Fig. 12c, d. Together, we changed the expression ‘wavenumber’ used for the y-axis of initial graphs to ‘optical frequency’ and unified the y-axes of newly added graphs in Supplementary Fig. 11 and 12 with ‘optical frequency.’

Location: *Supplementary Information*

“From the linear fitting results in Supplementary Fig. 13c, d, R-squared values of 0.99997 and 0.99987 are confirmed in each linear variable period for the up-chirp and down-chirp, respectively.”

Location: *Supplementary Information*

Supplementary Fig. 12. Linearity evaluation on flutter-wavelength modulation. (a) Raw interference signal; **(b)** Relative optical frequency behavior; Linear analysis including a regular residual plot of **(c)** up-chirp and **(d)** down-chirp.

<Comment 6>

What do the authors refer to with the measured linearity in line 269? Is it the linearity of velocity measurement? This should be a sine function not a linear function.

<Response>

We genuinely appreciate your detailed review. The mention of ‘linearity’ pertained to the R-squared value derived from a linear fit applied to data that exhibited apparent linearity across positive and negative velocities. Our intention was to convey the degree of correlation between

the measured and calculated theoretical velocities. However, we recognize that this assumption of linearity may have the potential to cause confusion among readers, as pointed out by the reviewer. Therefore, we have removed the inaccurate expression and revised the manuscript as follows:

Location: page 14, line 271-276

“To provide a more in-depth verification, we conducted an analysis of the velocity distribution along a single spatio-spectral axis of the 3D velocity image. In Fig. 3a, a linear velocity change transitions from positive to negative values, as expected from the alignment of the rotating disk. This velocity distribution closely follows the theoretical velocity, indicated by the black dash-dotted line (Fig. 3a), which is calculated relative to the propagating beam direction within the horizontal FOV³⁰.”

[30] Suyama, S., Ito, H., Kurahashi, R., Abe, H. & Baba, T. Doppler velocimeter and vibrometer FMCW LiDAR with Si photonic crystal beam scanner. *Opt Express* **29**, 30727 (2021).

<Comment 7>

All practical imaging experiments (see Fig 4.) are performed using a horizontal FOV substantially smaller than the value of 11.1° quoted in line 220. Do the authors find a degradation of the imaging quality and resolution at the edges of the FOV?

<Response>

Thank you for your meticulous and thorough review. We wish to address a preliminary matter regarding the field-of-view (FOV) specifications. The corrected FOV of 14.48° represents the maximum theoretical FOV attainable for a given spectral bandwidth of 160 nm, centered at 1535 nm, utilizing a transmission grating with a groove density of 1000 grooves/mm and an angle of incidence of 50°.

Fig. 7.1. Continuous wavelength sweep bandwidth of FWSL.

However, as shown in the following Fig. 7.1., the optical intensity of the FWSL tends to decrease as the wavelength moves to the edge of the spectral bandwidth. In addition, linewidth

broadening may occur at the edges. As the reduced power lowers the detection sensitivity and the broadened linewidth shortens the coherence length, which adversely affects the image quality, we restricted FOVs for all the images to the wavelength band where stable acquisition is feasible.

<Comment 8>

Please reconsider your choice of color map. The yellow to brown colors used to depict distances are very hard to distinguish for the reader. Fig 2(f) is unreadable. The velocity color maps is better but positive and negative values around "0" are also hard to distinguish for the reader as they are both shades of grey and purple. Please add axis labels for Figure 2, Panels (f) and (g).

<Response>

Thank you for your sincere comments and valuable suggestions.

Fig. 8.1 (a) Photograph of the rotating disk. The scanned area is depicted as a red box; (b) Theoretical distance of the scanned area of the disk according to the horizontal field of view (FOV); (c) three-dimensional (3D) distance image for the scanned area from the previous manuscript.

We agree that the color maps for both distance and velocity were not conducive to reading. As the reviewer has pointed out, the 3D distance image in initial Fig. 2f (Fig. 8.1c) from the previous manuscript, which was particularly difficult to distinguish, may have confused the readers. The 3D distance image did not appear in dynamically changing colors because the actual distance difference was within 0.6 cm⁶, as shown in Fig. 8.1b, which is a theoretical distance that corresponds to the scanner area.

After careful deliberation, we have come to the conclusion that the examination of such distance measurement results may not hold significant relevance in verifying our proposed system. Consequently, we have made the decision to remove the 3D distance image and its associated analysis from our presentation. Instead, we have redirected our focus on providing a comprehensive interpretation of the velocity measurement, as outlined below.

[6] Suyama, S., Ito, H., Kurahashi, R., Abe, H. & Baba, T. Doppler velocimeter and vibrometer FMCW LiDAR with Si photonic crystal beam scanner. *Opt Express* **29**, 30727 (2021).

Location: page 14-15, line 271-298

“As a result, we successfully obtained a clear 3D velocity image, as illustrated in Fig. 2f. To provide a more in-depth verification, we conducted an analysis of the velocity distribution along a single spatio-spectral axis of the 3D velocity image. In Fig. 3a, a linear velocity change transitions from positive to negative values, as expected from the alignment of the rotating disk. This velocity distribution closely follows the theoretical velocity, indicated by the black dash-dotted line (Fig. 3a), which is calculated relative to the propagating beam direction within the horizontal FOV³⁰. For comprehensive calculations, please refer to the Supplementary Information.

In Fig. 3b, we present error values, all within magnitudes not exceeding 0.08 m/s across the spatio-spectral axis. These errors may have originated from unintended vibrations of the disk and the inherent ellipticity of the disk. Nevertheless, it is evident that the measured velocities remain reliable within a reasonable error rate of around 5%, except for the zero velocity part.

To further demonstrate the precision of our system’s velocity measurement across the spatio-spectral axis, we provided a standard deviation (SD) of 100 measured values at each lateral point, as shown in Fig. 3c. Notably, the SD is recorded consistently within the 0.1 m/s range throughout the spectral scanning spanning from 1549 nm to 1564 nm. A few spatio-spectral points exhibit relatively high SDs, but they do not exceed 0.2 m/s. Considering the native velocity resolution determined by the FWSL, the obtained velocity results stand within a reasonably accurate range.

We selected three representative points each for positive, negative, and zero velocity parts among the different spatio-spectral points for further investigation on the precision. The histograms of measured velocity during 100 measurements at three selected wavelengths of 1544.7 nm, 1554.0 nm, and 1563.8 nm are provided in the inset of Fig. 3c. It is evident that a clear tendency of a normal distribution is observed at all the points. The mean error values of the estimated normal distribution at each point are recorded as 0.02 m/s, 0.03 m/s, and 0.003 m/s, respectively, representing our experimental velocity measurement accuracy.

Corresponding SDs are 0.063 m/s, 0.052 m/s, and 0.062 m/s, respectively, indicating that the actual velocity measurement is more precise than the theoretical velocity resolution. We suggest that the proposed system successfully performed a precise velocity measurement during spatio-spectral scanning, showing the unique strength of the proposed FWSL. This achievement is the first time, as of now, for the spatio-spectral coherent LiDAR based on a wavelength-swept laser.”

[30] Suyama, S., Ito, H., Kurahashi, R., Abe, H. & Baba, T. Doppler velocimeter and vibrometer FMCW LiDAR with Si photonic crystal beam scanner. *Opt Express* **29**, 30727 (2021).

Together, we accepted the reviewer's suggestion and revised the color maps. The updates feature more contrasting colors, as illustrated below.

Location: page 12, line 242

Fig. 2 | Characterization of FWSL and coherent LiDAR system.

Location: page 15, line 300

Fig. 3 | Velocity measurement accuracy and precision.

Fig. 4 | 4D coherent ranging results.

[Reviewer 3]

This work aims to perform a solid-state coherent LiDAR. The authors propose a novel flutter-wavelength-swept laser that offers a simultaneous yet independent wavelength modulation of 5–6 pm for axial distance ranging and velocity measurement, and a wavelength sweep of 160 nm for horizontal beam scanning. However, there exists a severe concern that the experiment is limited and lacks enough evidence for technique validation. The following concerns should be carefully addressed.

<Comment 1>

The introduction could benefit from a more detailed explanation of the performance and immaturity of silicon chip-based technology in relation to Coherent LiDAR. It would be helpful to clarify the limitations of this technology when compared to scanning-based Coherent LiDAR, allowing readers to better understand the research's contribution to the field.

<Response>

We appreciate reviewer for your meticulous review of our manuscript. We agree that information regarding the coherent light detection and ranging (LiDAR) technology based on silicon chips is insufficient. Therefore, we added a more comprehensive explanation of silicon chip-based coherent LiDAR technology, and revised the manuscript as follows:

Location: page 3-4, line 41-55

“In addition to its strengths of coherent measurement, there are ongoing initiatives to develop high-performance coherent LiDAR systems that excel in terms of accuracy, speed, resolution, range, stability, efficiency, and other critical aspects. Various approaches have been reported to achieve higher performance, especially in solid-state systems¹⁵. Flash LiDAR^{4,16,17} exhibits a powerful performance of 3D distance and velocity detection at long range with a compact size of on-chip architecture. However, it does have a drawback, which is its limited number of pixels. The resulting narrow field of view (FOV) and the consequent requirement for free-space optics indicate that the technology still requires further maturation. Similarly, LiDAR systems based on the optical phased array (OPA)¹⁸⁻²⁰ deliver robust performance with silicon chip architecture and offer a wider FOV without relying on a lens. However, the fabrication of OPA involves a challenging trade-off between the FOV, SNR, and beam efficiency. Additionally, as of now, a full four-dimensional (4D) image of both distance and velocity measurements has not been reported using this technology. Coherent LiDAR, which uses spectral deflection^{4,21-24}, is a strong alternative, especially when compared to the existing state of silicon chip-based LiDARs. Spatio-spectral mapping simplifies and makes it straightforward to create a direct correspondence of the lateral position of the beam according to wavelength sweep, all driven by a single light source.”

[4] Rogers, C. *et al.* A universal 3D imaging sensor on a silicon photonics platform. *Nature* **590**, 256–261 (2021).

- [15] Li, N. *et al.* A Progress Review on Solid-State LiDAR and Nanophotonics-Based LiDAR Sensors. *Laser Photon Rev* **16**, (2022).
- [16] Martin, A. *et al.* Photonic integrated circuit-based FMCW coherent LiDAR. *Journal of Lightwave Technology* **36**, 4640–4645 (2018).
- [17] Baba, T. *et al.* Silicon Photonics FMCW LiDAR Chip With a Slow-Light Grating Beam Scanner. *IEEE Journal of Selected Topics in Quantum Electronics* **28**, (2022).
- [18] Poulton, C. V. *et al.* Coherent solid-state LIDAR with silicon photonic optical phased arrays. *Opt Lett* **42**, 4091 (2017).
- [19] Poulton, C. V. *et al.* Long-Range LiDAR and Free-Space Data Communication With High-Performance Optical Phased Arrays. *IEEE Journal of Selected Topics in Quantum Electronics* **25**, 1–8 (2019).
- [20] Bhargava, P. *et al.* Fully Integrated Coherent LiDAR in 3D-Integrated Silicon Photonics/65nm CMOS. in *2019 Symposium on VLSI Circuits C262–C263* (IEEE, 2019). doi:10.23919/VLSIC.2019.8778154.
- [21] Okano, M. & Chong, C. Swept Source Lidar: simultaneous FMCW ranging and nonmechanical beam steering with a wideband swept source. *Opt Express* **28**, 23898 (2020).
- [22] Li, Z., Zang, Z., Han, Y., Wu, L. & Fu, H. Y. Solid-state FMCW LiDAR with two-dimensional spectral scanning using a virtually imaged phased array. *Opt Express* **29**, 16547 (2021).
- [23] Lukashchuk, A., Riemensberger, J., Karpov, M., Liu, J. & Kippenberg, T. J. Dual chirped microcomb based parallel ranging at megapixel-line rates. *Nat Commun* **13**, 3280 (2022).
- [24] Qian, R. *et al.* Video-rate high-precision time-frequency multiplexed 3D coherent ranging. *Nat Commun* **13**, (2022).

<Comment 2>

To demonstrate the effectiveness of your method in comparison to conventional swept LiDAR, it would be useful to include additional quantitative results that showcase differences in distance and velocity accuracy. You should provide these quantitative comparative results compared with other LiDAR methods in the manuscript.

<Response>

We appreciate the reviewer for detailed review with valuable suggestions. We align with the reviewer's recommendation, and, in response, we have included a table in the Supplementary Information that summarizes quantitative specifications, encompassing distance and velocity resolution. Supplementary Table 1 below now offers a comprehensive and effective comparison between our system and other spatio-spectral coherent LiDARs.

Location: *Supplementary Information*

Supplementary Table 1. Specifications of spatio-spectral coherent LiDAR.

Content	Laser specification						
	Coherence length	Center Wavelength (nm)	Sweep bandwidth (THz)	Sampling point interval (nm/pt)	Axial resolution (cm)	Acquisition rate (MHz)	Fast axis scan rate (kHz)
Okano et al. ¹	> 150 m (HSL-1, Santec)	1060	11	1.13	0.042	0.3 (N=30)	10
			5	0.412	0.14	0.013 (N=45)	0.3
Qian et al. ²	> 1 m (akinetic allsemiconductor programmable swept laser, Insight Photonics Solutions)	1316	11.4	0.27	0.282	7.6 (N=475)	15.94
Ours (Flutter-wavelength modulation)	> 1.18 km	1535	11.2 ^①	0.0056 ^②	21	0.1 ^④	2 ^③
Ours (Conventional wavelength sweep)	> 1.18 km	1535	11.2 ^①	0.44 ^②	0.27	0.4 ^④ (N=200)	2 ^③
Content	LiDAR system specification						
	Electric bandwidth		Maximum measurable distance (m)	Distance resolution (cm)	Velocity resolution (cm/s)	Frame rate (Hz)	
Photodetector (GHz)	Digitizer (GS/s)						
Okano et al. ¹	1	1	0.25 (*linear tuning period : 35 μ s)	N.A.	N.A.	100 (30x100 px)	
			12 (*linear tuning period : 800 μ s)	N.A.	N.A.	1.5 (45x200 px)	
Qian et al. ²	0.4	0.8	0.33	< 0.08	N.A.	33.2 (475x400 px)	
Ours (Flutter-wavelength modulation)	0.4	0.5	264	< 4	< 20	10 ^⑦ (200 ^⑥ x45 ^⑤ px)	
Ours (Conventional wavelength sweep)	0.4	0.5	1.7	< 4	< 20	44.4 ^⑦ (200 ^⑤ x45 ^⑥ px)	

<Comment 3>

When considering the motion of the LiDAR in your method, it is important to address how data collection and any jitter during the process are managed. Elaborating on this aspect will provide greater insight into the system's functionality under various conditions.

<Response>

Thank you for your insightful suggestion. One of the most important concerns in the practical application of LiDAR systems lies in addressing artifacts stemming from motion. Conventional LiDAR systems based on mechanical scanners are especially vulnerable to external vibrations and can easily generate jitter, which can cause additional motion artifacts.

Our proposed system employing a solid-state scanner was robust against defects caused by internal or external motion. Moreover, the scanning mechanisms for both axial and lateral directions of our proposed system are electrically controlled in the triggered state to ensure synchronized operation and data acquisition, regardless of the external turbulence or noise frequency. In this regard, we expect the system to perform robustly against mechanical movements.

Taking into consideration the suggestion of the reviewer, we have made the following revisions to our manuscript.

Location: page 21, line 418-421

“In addition, the AOD is controlled electrically in the triggered state with the EOPM and WS, which are also controlled electrically. The electrical based on the synchronized system, which especially operates in a solid state, ensures robust data acquisition regardless of external turbulence or optical jitter.”

<Comment 4>

While the article mentions the use of a spectral domain method for the LiDAR technique, it is necessary to discuss its performance in challenging environments such as rain, fog, and areas with pollution or smoke. Including further explanation and data-supported evidence for these scenarios will ensure a more comprehensive evaluation of the method's effectiveness across different conditions.

<Response>

We appreciate the reviewer for the careful comments and suggestions. We admit that demonstration of our system performance in challenging environments is necessary, considering the powerful features of coherent LiDAR. In response to this valuable feedback, we conducted additional coherent measurements under an artificially simulated foggy environment using our system. We performed four-dimensional (4D) real-time imaging on a simulated foggy environment, and corresponding results are provided in Supplementary Video 2. It is obvious that the three-dimensional (3D) distance and velocity of both stationary and moving objects are accurately measured simultaneously regardless of the environment, comparing the 100% transmittance situation (Supplementary Fig. 16a) and the low transmittance situation (Supplementary Fig. 16b-c). We suggest that the results further demonstrate the robustness of the proposed LiDAR system relative to its powerful and effective 4D imaging.

We have also taken the initiative to provide an elaborate account of this experiment in a newly added section within the Supplementary Information as below:

Location: *Supplementary Information*

“4D coherent ranging under an artificial fog environment

Scene D in Supplementary Fig. 16 was newly designed for real-time 4D imaging verification in a simulated foggy environment. Against a retroreflective box at 4.4 m in the background, the rail was located at 3.5 m, and the tree was located at 3.8 m, which are the same objects used in Scenes A and C. To create an artificial fog environment, a studio that could cover the rail was established, and an air distributor was installed on the ceiling of the studio to uniformly spray the smog. A photodiode sensor was placed in the background to measure the transmittance. The transmittance was calculated by detecting the optical power of the 1550 nm

center wavelength reference laser light passing through the studio. A fiber collimator used to couple the reference laser light into free space was located parallel to our optical system, and the collimated beam was aligned in a straightforward manner to the photodiode sensor. The ROI in Supplementary Fig. 16a was imaged at 200×45 pixels in real time under the scanning conditions of $6.9^\circ \times 1.63^\circ$ (H \times V) with an acquisition rate of 100 kHz and a fast axis scan rate of 2 kHz. The final output power of the system was boosted to 20 mW using a booster optical amplifier (Thorlabs, Newton, NJ, USA). A full video of the measurements with photographs and the transmittance of the studio environment is shown in Supplementary Video 2. It is obvious that the 3D distance and velocity of both stationary and moving objects are accurately measured simultaneously regardless of the environment, comparing the 100% transmittance situation (Supplementary Fig. 16a) and the low transmittance situation (Supplementary Fig. 16b-c). We suggest that the results further demonstrate the robustness of the proposed LiDAR system relative to its powerful and effective 4D imaging.

Supplementary Fig. 16. Real-time 4D coherent ranging under a simulated foggy environment. Photograph of the studio environment, 4D distance and velocity image of Scene D imaged with a pixel size of 200×45 and a frame rate of 10 Hz (Lower left side), and transmittance at a given environment (Lower right side). Four individual frames were captured from the full video of 14.8 s, provided in Supplementary Video 2.”

<Comment 5>

Although the images in the article are visually appealing, it would be beneficial to include a photograph of an actual experimental setup, showcasing each device and its placement. This addition will help readers gain a clearer understanding of the experiment's execution and enhance the overall presentation of the research.

<Response>

We sincerely appreciate your valuable comments and accept the reviewer's suggestion. We have included a photograph of our proposed experimental setup and of an overall actual measurement scene in the Supplementary Information below:

Location: *Supplementary Information*

“

Supplementary Fig. 18. Photograph of the experimental setup.

Supplementary Fig. 19. Photograph of the actual measurement scene during the real-time 4D coherent ranging under a simulated foggy environment.”

REVIEWER COMMENTS

Reviewer #1 (Remarks to the Author):

I would like to thank the authors for carefully addressing my comments. The authors have presented the structure and operational principle of the Flutter-Wavelength-Modulated Lidar (FWML). However, it may not be immediately clear what sets their work apart from previous research. The same function may also be obtained with integrated external cavity lasers as demonstrated in [1-2].

Also, the claim made in line 53 on page 3 appears to be problematic. It is not clear what the term "spectral deflection" precisely refers to, and it raises questions regarding whether the author is excluding OPA from the category of spectral deflection methods or excluding OPA from the silicon chip-based Lidars. It is essential for the author to conduct a thorough and well-informed investigation before making such claims.

Furthermore, the main challenge of coherent LiDAR today is still the low acquisition rate of point cloud. A low acquisition rate of point cloud will lead to a decrease in the number of point cloud distributions on the surface of distant targets, thus causing blind spots in the field of view (FOV) and resulting in objects being 'unseen'. This could significantly affect the safety of autonomous driving in terms of road cruising functions. This is why some commercial LiDAR companies' related products (e.g., Innovusion's Falcon series) use localized encrypted point cloud scanning to increase the density of the point cloud in the region of interest (ROI), which is essentially a compromise to cope with the contradiction between the demand for high imaging resolution and the insufficient point cloud acquisition rate. Alternatively, higher point cloud densities can be achieved by using a lower FOV range, as demonstrated by the authors in the 3D imaging section (page 17, line 336) of this manuscript. Again, this is a compromise strategy due to the insufficient acquisition rate of the point cloud. Parallel detection or multi-line LiDARs may offer advantages for solving this challenge, but I don't think the current light source used in this work has surpassed advantages in size and integration capability, compared with the well-reported integrated external cavity lasers. So, the authors need to provide a clearer picture of the novelty of their work, and a more comprehensive discussion and comparison between the demonstrated light source with other various light sources should be made. As stated in line 100 on page 5, the light source plays the most crucial role in the development of a spatio-spectral coherent Lidar system.

It should also be noted that the spatio-spectral architecture mentioned in the manuscript has two obvious drawbacks, which were not discussed. First, there is a trade-off between the lateral and vertical point density, because the point cloud acquisition rate is fixed, and the system uses a sequenced scanning process. Therefore, increasing the lateral point density will inevitably sacrifice the vertical resolution. Second, the horizontal scan only requires one chirp cycle to complete the target point detection, which is unrealistic in practice, and typically more cycles are needed to improve the detection success rate.

Based on the above comments, I don't think this manuscript can be published in Nature communications.

Minor comments:

In Figure 3.1, could you please clarify how the noise floor is determined? It appears that there may be instances where background noise fluctuates above the indicated noise floor.

[1] Liu C, Xu R, Xu W, et al. High-precision FMCW ranging with a hybrid-integrated external cavity laser[C]//2023 Opto-Electronics and Communications Conference (OECC). IEEE, 2023: 1-4.

[2] Li M, Chang L, Wu L, et al. Integrated pockels laser[J]. Nature communications, 2022, 13(1): 5344.

Reviewer #2 (Remarks to the Author):

I strongly recommend the revised version of the manuscript for publication in Nature Communication and commend the authors for following up on all the questions regarding the laser technology and the measurement methodology so thoroughly.

Reviewer #3 (Remarks to the Author):

The authors have addressed the reviewer's concerns. The manuscript can be forwarded to publication.

Response to Reviewers' comments

Manuscript Number: NCOMMS-23-08645A

Title: Spatio-spectral 4D Coherent Ranging Using a Flutter-wavelength-swept Laser

Authors: Dawoon Jeong, Hansol Jang, Min Uk Jung, Taeho Jeong, Hyunsoo Kim, Sanghyeok Yang, Janghyeon Lee, and Chang-Seok Kim

[Reviewer 1]

I would like to thank the authors for carefully addressing my comments. The authors have presented the structure and operational principle of the Flutter-Wavelength-Modulated Lidar (FWML). However, it may not be immediately clear what sets their work apart from previous research. The same function may also be obtained with integrated external cavity lasers as demonstrated in [1-2].

<Response>

We thank the reviewer for your meticulous and valuable review. Considering the reviewer's concerns, we investigated previous studies on lasers, including the two papers that the reviewer has attached. First, the conventional free space cavity-based external cavity lasers (ECLs) have been developed with a focus on mode-hop-free, unidirectional wavelength tuning over a wide spectral bandwidth since their main purpose was spectroscopic application¹⁻⁹. Among them, there was an ECL of a structure with electro-optically controllable materials inserted into the cavity, but it also appeared to be able to perform only one function between fine wavelength modulation and coarse wavelength sweep¹⁰⁻¹⁴. Meanwhile, tunable diode laser absorption spectroscopy is a technology that utilizes both wavelength scanning and modulation of ECL simultaneously. However, the modulation shape is different from the proposed flutter-wavelength-swept laser (FWSL), and the purpose is completely different in that it is for spectroscopic applications¹⁵, not for distance ranging.

Next, like the reference papers^{16,17} mentioned by the reviewer, hybrid ECLs integrated into a photonic integrated circuit (PIC) are also being actively studied for various purposes including frequency-modulated continuous wave (FMCW) light detection and ranging (LiDAR)¹⁸. Various hybrid ECLs based on silicon (Si) and silicon nitride (Si₃N₄) have been reported which provide a wide tuning bandwidth in a compact size¹⁶⁻¹⁹. However, they have limitations of slow speed (few kHz modulation speed) and low power efficiency because they perform wavelength tuning based on the thermo-optic effect. Therefore, they cannot support 100 kHz flutter-wavelength modulation during continuous wavelength sweep for a given spectral bandwidth like FWSL¹⁹. Due to the speed limitations of Si-based hybrid ECL, a Pockels laser based on a lithium niobate (LN) has been proposed, which supports electro-optic-based fast wavelength modulation¹⁷. However, coarse wavelength tuning still relies on the thermo-optic effect, and it has only shown spectral bandwidth through discrete wavelength tuning and fast switching between only two lasing modes, respectively. That is, there has not been verification of simultaneous fine flutter-wavelength modulation and coarse wavelength sweep across the entire spectral bandwidth in a synchronized state like the proposed FWSL.

In these reasons, we propose that FWSL is the only laser source among the lasers reported to date that ensures a narrow linewidth and supports both fast, linear flutter-wavelength modulation and continuous, wide-bandwidth wavelength sweep at the same time.

[1] Zhu, Y., Liu, Z., Zhang, X., Shao, S. & Yan, H. Dynamic mode matching of internal and external cavities for enhancing the mode-hop-free synchronous tuning characteristics of an external-cavity diode laser. *Applied Physics B* **125**, 217 (2019).

- [2] Wysocki, G. *et al.* Widely tunable mode-hop free external cavity quantum cascade laser for high resolution spectroscopic applications. *Applied Physics B* **81**, 769–777 (2005).
- [3] Shin, D. K. *et al.* Widely tunable, narrow linewidth external-cavity gain chip laser for spectroscopy between 10 – 11 μm . *Opt Express* **24**, 27403 (2016).
- [4] Foster, S., Cranch, G. A. & Tikhomirov, A. Experimental evidence for the thermal origin of $1/f$ frequency noise in erbium-doped fiber lasers. *Phys Rev A (Coll Park)* **79**, 053802 (2009).
- [5] Repasky, K. S., Nehrir, A. R., Hawthorne, J. T., Switzer, G. W. & Carlsten, J. L. Extending the continuous tuning range of an external-cavity diode laser. *Appl Opt* **45**, 9013 (2006).
- [6] Gong, H., Liu, Z., Zhou, Y. & Zhang, W. Extending the mode-hop-free tuning range of an external-cavity diode laser by synchronous tuning with mode matching. *Appl Opt* **53**, 7878 (2014).
- [7] Gong, H., Liu, Z., Zhou, Y., Zhang, W. & Lv, T. Mode-hopping suppression of external cavity diode laser by mode matching. *Appl Opt* **53**, 694 (2014).
- [8] Dutta, S., Elliott, D. S. & Chen, Y. P. Mode-hop-free tuning over 135 GHz of external cavity diode lasers without antireflection coating. *Applied Physics B* **106**, 629–633 (2012).
- [9] Ménager, L., Cabaret, L., Lorgeré, I. & Le Gouët, J.-L. Diode laser extended cavity for broad-range fast ramping. *Opt Lett* **25**, 1246 (2000).
- [10] Laschek, M., Wandt, D., Tünnermann, A. & Welling, H. Electro-optical frequency modulation of an external-cavity diode laser. *Opt Commun* **153**, 59–62 (1998).
- [11] Levin, L. Mode-hop-free electro-optically tuned diode laser. *Opt Lett* **27**, 237 (2002).
- [12] Chen, C.-Y. Fine-tuning of a diode laser wavelength with a liquid crystal intracavity element. *Optical Engineering* **43**, 234 (2004).
- [13] Wang, P., Seah, L. K., Murukeshan, V. M. & Chao, Z. External-cavity wavelength tunable laser with an electro-optic deflector. *Appl Opt* **45**, 8772 (2006).
- [14] Shen, L., Ye, Q., Cai, H. & Qu, R. Mode-hop-free electro-optically tuned external-cavity diode laser using volume Bragg grating and PLZT ceramic. *Opt Express* **19**, 17244 (2011).
- [15] Avetisov, V., Bjoroey, O., Wang, J., Geiser, P. & Paulsen, K. G. Hydrogen Sensor Based on Tunable Diode Laser Absorption Spectroscopy. *Sensors* **19**, 5313 (2019).
- [16] Liu, C. *et al.* High-precision FMCW ranging with a hybrid-integrated external cavity laser. in *2023 Opto-Electronics and Communications Conference (OECC)* 1–4 (IEEE, 2023). doi:10.1109/OECC56963.2023.10209936.
- [17] Li, M. *et al.* Integrated Pockels laser. *Nat Commun* **13**, 5344 (2022).
- [18] Hu, M., Pang, Y. & Gao, L. Advances in Silicon-Based Integrated Lidar. *Sensors* **23**, 5920 (2023).

[19] Porter, C., Zeng, S., Zhao, X. & Zhu, L. Hybrid integrated chip-scale laser systems. *APL Photonics* **8**, (2023).

Also, the claim made in line 53 on page 3 appears to be problematic. It is not clear what the term "spectral deflection" precisely refers to, and it raises questions regarding whether the author is excluding OPA from the category of spectral deflection methods or excluding OPA from the silicon chip-based Lidars. It is essential for the author to conduct a thorough and well-informed investigation before making such claims.

<Response>

We appreciate your review and the important clarification you've raised. We acknowledge that the expression spectral deflection in the sentence pointed out by the reviewer may have confused readers as to its meaning. Spectral deflection was used to represent the beam deflection according to the spectral information of light. However, we replaced spectral deflection with spectral dispersion, which conveys the corresponding meaning more accurately. In addition, we would like to mention that optical phased array (OPA)-based LiDAR is classified and designated as silicon chip-based LiDAR together with flash LiDAR in the sentence since OPA is fabricated based on a silicon chip platform¹⁸.

Location: page 3, line 52-54

"Coherent LiDAR using spectral dispersion^{4,21-24} is a strong alternative, especially when compared to the existing state of silicon chip-based LiDARs, which includes flash and OPA-based LiDAR."

Furthermore, the main challenge of coherent LiDAR today is still the low acquisition rate of point cloud. A low acquisition rate of point cloud will lead to a decrease in the number of point cloud distributions on the surface of distant targets, thus causing blind spots in the field of view (FOV) and resulting in objects being 'unseen'. This could significantly affect the safety of autonomous driving in terms of road cruising functions. This is why some commercial LiDAR companies' related products (e.g., Innovusion's Falcon series) use localized encrypted point cloud scanning to increase the density of the point cloud in the region of interest (ROI), which is essentially a compromise to cope with the contradiction between the demand for high imaging resolution and the insufficient point cloud acquisition rate. Alternatively, higher point cloud densities can be achieved by using a lower FOV range, as demonstrated by the authors in the 3D imaging section (page 17, line 336) of this manuscript. Again, this is a compromise strategy due to the insufficient acquisition rate of the point cloud. Parallel detection or multi-line LiDARs may offer advantages for solving this challenge, but I don't think the current light source used in this work has surpassed advantages in size and integration capability, compared with the well-reported integrated external cavity lasers. So, the authors need to provide a clearer picture of the novelty of their work, and a more comprehensive discussion and comparison between the demonstrated light source with other various light sources should be made. As

stated in line 100 on page 5, the light source plays the most crucial role in the development of a spatio-spectral coherent Lidar system.

<Response>

We thank the reviewer for your careful review and the important suggestions. As the reviewer mentioned, the fast acquisition rate of the system for sufficient point cloud density is a crucial specification in LiDAR applications. To enhance the low acquisition rate of coherent LiDAR, there has been a novel strategy of utilizing multi-wavelength light sources^{20,21}. In previously reported research, the acquisition rate has reached up to a few MHz based on the multi-comb generation and parallel detection of each simultaneously modulated comb.

Fig. 1 Wavelength spectrum of flutter-wavelength-swept laser (FWSL) with a multi-channel wavelength selector (WS).

Fig. 1 shows the output spectrum shape that appears when the single-channel wavelength selector (WS), the existing WS, is replaced with a multi-channel WS in the FWSL. Here, the multi-channel WS is implemented by replacing the diffraction grating component with the Fabry-Perot etalon component. In this case, simultaneous flutter-wavelength modulation through an electro-optic phase modulator (EOPM) is possible for all transmission wavelengths of the longitudinal mode of the external cavity. The result suggests that FWSL is also capable of multi-wavelength generation and simultaneous modulation of multiple wavelengths through simple structural modification. Therefore, we propose that FWSL is a promising light source with a feasible variety that can implement both sequential and parallel coherent LiDAR based on single and multi flutter-wavelength modulation, respectively.

Fig. 2 Raw interference signal by a single electro-optic phase modulator (EOPM) modulation of 10 MHz.

Meanwhile, in the case of the present sequential coherent LiDAR based on the FWSL with the single-channel WS, there is room for an acquisition rate increase of more than a hundred times. Fig. 2 shows an interference signal obtained from a fiber-based Mach-Zehnder interferometer when EOPM is modulated with a 10 MHz sine waveform at the center wavelength of 1535 nm. Clear interference signal proves that flutter-wavelength modulation of FWSL functions well even in a high-speed modulation condition of 10 MHz. This result suggests that the optimal modulation condition, which is limited by the current equipment such as an arbitrary function generator or a voltage amplifier, was simply set to 100 kHz, and not a limitation of the FWSL itself. In addition, the spatio-spectral scanning speed can be increased by applying a high-speed electro-optic deflector such as potassium tantalate niobate (KTN) to single-channel WS. Previous research that demonstrated fast wavelength tuning (tens to hundreds of kHz) over a wide bandwidth (tens to a hundred nm) using KTN^{22,23} suggests a possibility that FWSL can achieve a high spatio-spectral scanning speed of hundreds of kHz level. Since higher spatio-spectral scanning speed ensures a higher frame rate of the proposed LiDAR system, we expect that the FWSL-based sequential coherent LiDAR of a hundred times higher acquisition rate can be achieved to secure sufficient point cloud density.

However, considering the nature of FMCW LiDAR, which utilizes a frequency-modulated signal with periodicity, the distance that can be measured without ambiguity is not only affected by coherence length but also determined by the modulation speed. When the wavelength is modulated using a 50% symmetry triangle waveform, the ambiguous distance is determined as

$$D_{amb} = \frac{c}{2f_{sweep}} \quad (1)$$

where c and f_{sweep} refer to the speed of light in vacuum and the flutter-wavelength modulation rate, respectively.

Fig. 3 (a)-(d) Principle of ambiguous distance formation as the optical path difference increases; **(e)-(h)** Beat frequency, f_b , change aspects based on the distance change ($d_2 > d_1$) at each condition.

Fig. 3 shows the beat frequency (f_b) formation and change aspects for each distance condition of the measurement target. First, Fig. 3a, e shows the formation of f_b in the area where distance can be normally measured ($0 \leq d < D_{amb}$). Both the reference signal (S_{ref}) and the reflected signal (S_{sig}) that form f_b in the corresponding area are of the same order, and therefore, as the distance increases ($d_1 \rightarrow d_2$), f_b also increases proportionally. Fig. 3b and f shows the formation of f_b when the position of the target is equal to the ambiguous distance ($d = D_{amb}$). As shown in the figure, f_b is not measured because there is no simultaneously overlapping area between chirping signals in the same direction, f_b is not measured, as seen from the figure. Fig. 3c, g illustrates the case when the condition $0 \leq d < 2D_{amb}$ holds. In this case, $S_{sig,N}$ forms an interference signal with $S_{ref,N+1}$, which is the next-order reference signal and f_b decreases as distance increases. This phenomenon can be overcome using complex conjugate resolutions²⁴. Finally, Fig. 3d and h show that f_b increases proportionally as the distance increases; however, when the order (N) between the reference and the reflected signals differs by 1, it can be observed that Fig. 3d and h have the same shape as Fig. 3a and e because of ambiguity. Therefore, an optimal modulation speed should be considered to prevent such occurrences of ambiguity in every FMCW LiDAR.

As the reviewer mentioned, PIC-based laser is a promising research field that is being actively explored. Considering the compact and powerful use of PIC-based lasers, the integration of the proposed FWSL would be one of the meaningful development directions for further study. As various approaches for specifications that are essential to implement the performance of FWSL on PIC are being proposed^{17,25}, we expect that the integration of the FWSL will be feasible in the future. In this regard, we revised our discussion as below.

[20] Riemensberger, J. *et al.* Massively parallel coherent laser ranging using a soliton microcomb. *Nature* **581**, 164–170 (2020).

[21] Lukashchuk, A., Riemensberger, J., Karpov, M., Liu, J. & Kippenberg, T. J. Dual chirped microcomb based parallel ranging at megapixel-line rates. *Nat Commun* **13**, 3280 (2022).

[22] Ling, Y., Yao, X. & Hendon, C. P. Highly phase-stable 200 kHz swept-source optical coherence tomography based on KTN electro-optic deflector. *Biomed Opt Express* **8**, 3687 (2017).

[23] Fujimoto, M. *et al.* Stable wavelength-swept light source designed for industrial applications using KTN beam-scanning technology. in *Photonic Instrumentation Engineering IV* (eds. Soskind, Y. G. & Olson, C.) vol. 10110 101100Q (SPIE, 2017).

[24] Sarunic, M. V., Choma, M. A., Yang, C. & Izatt, J. A. Instantaneous complex conjugate resolved spectral domain and swept-source OCT using 3x3 fiber couplers. *Opt Express* **13**, 957 (2005).

[25] Sarunic, M. V., Choma, M. A., Yang, C. & Izatt, J. A. Instantaneous complex conjugate resolved spectral domain and swept-source OCT using 3x3 fiber couplers. *Opt Express* **13**, 957 (2005).

Location: page 23, line 457-461

“Additionally, in consideration of the future development of coherent LiDAR, we expect that the proposed FWSL can branch into various future studies. As faster and more compact light sources and LiDAR systems are preferred, an approach on parallel LiDAR based on simultaneous multi-wavelength generation^{5,23}, or miniaturization of the FWSL through integration such as PIC-based lasers³⁸ will be meaningful further studies.”

It should also be noted that the spatio-spectral architecture mentioned in the manuscript has two obvious drawbacks, which were not discussed. First, there is a trade-off between the lateral and vertical point density, because the point cloud acquisition rate is fixed, and the system uses a sequenced scanning process. Therefore, increasing the lateral point density will inevitably sacrifice the vertical resolution. Second, the horizontal scan only requires one chirp cycle to complete the target point detection, which is unrealistic in practice, and typically more cycles are needed to improve the detection success rate.

Based on the above comments, I don't think this manuscript can be published in Nature communications.

<Response>

We appreciate the reviewer for your detailed review. First, the trade-off between the lateral and vertical point density can also be solved by both methods mentioned above. FWSL can function for parallel LiDAR by adopting multi-channel WS, which can solve the trade-off by simultaneously detecting multiple channels along a single lateral axis. In addition, the bottleneck of lateral point density can be resolved by increasing the speed of EOPM to achieve a hundred times higher acquisition rate.

Second, if the statement that 'horizontal scanning should be more than one cycle' means that multiple frames should be averaged to detect the target in practice, we agree with the reviewer. Throughout the study, we conducted experiments with a focus on proposing the FWSL and demonstrating solid-state four-dimensional (4D) imaging using the FWSL. In the process, we adopted the common unidirectional raster scanning technique, with acousto-optic scanning on the fast axis and spectral scanning on the slow axis. However, the non-mechanical scanner has the versatility to implement various scanning techniques. We expect that the detection success rate can be easily improved by introducing a bidirectional raster scanning technique and a frame averaging with a higher frame rate.

Lastly, while the reviewer's meticulous review and valuable suggestions have helped us to improve our study, we focused on addressing the limitations of spatio-spectral coherent LiDAR based on conventional wavelength-swept lasers by newly proposing the FWSL that decouples flutter-wavelength modulation and wavelength sweep. In addition, we suggest that no laser source has been reported yet that can function like the novel characteristics of FWSL based on our investigation above. The attached results have indeed successfully demonstrated the additional properties of our laser, as pointed out by the reviewer. However, we keep the manuscript to primarily emphasize proving the underlying principles and presenting the relevant outcomes of the synchronized simultaneous flutter-wavelength modulation and wavelength sweep in order to enhance the reader's comprehension of our study.

<Comment 1>

In Figure 3.1, could you please clarify how the noise floor is determined? It appears that there may be instances where background noise fluctuates above the indicated noise floor.

<Response>

Fig. 4 Fast Fourier transform (FFT) result of the interference signal by a single flutter-wavelength modulation. I , $\overline{I_{noise}}$, and σ_{noise} depicts the pure signal intensity, the average noise level, and the standard deviation of the noise intensity, respectively.

We thank the reviewer for the insightful questions you've raised. The noise floor in Fig. 4 is characterized by the average intensity of the noise signal, as generally defined by the noise floor in the depth profile of optical coherence tomography (OCT)²⁶. The measured intensity signal consists of the pure signal intensity, I in Fig. 4, biased by the average noise level, $\overline{I_{noise}}$ in Fig. 4. The derived signal-to-noise ratio (SNR) was about 42 dB with the measured peak signal intensity of 109 dB and the average noise level of 67 dB.

[26] Baumann, B. *et al.* Signal averaging improves signal-to-noise in OCT images: But which approach works best, and when? *Biomed Opt Express* **10**, 5755 (2019).

[Reviewer 2]

I strongly recommend the revised version of the manuscript for publication in Nature Communication and commend the authors for following up on all the questions regarding the laser technology and the measurement methodology so thoroughly.

[Reviewer 3]

The authors have addressed the reviewer's concerns. The manuscript can be forwarded to publication.

REVIEWERS' COMMENTS

Reviewer #1 (Remarks to the Author):

With the revision made in the manuscript, it's now acceptable for publication. I do not necessary agree with all the claims made, but this paper is technically sound.